# Mobilizing Personalized Federated Learning in Infrastructure-Less and Heterogeneous Environments via Random Walk Stochastic ADMM

**Ziba Parsons**
CIS Department
University of Michigan
Dearborn, MI
zibapars@umich.edu

**Fei Dou**
School of Computing
University of Georgia
Athens, GA
fei.dou@uga.edu

**Houyi Du**
CIS Department
University of Michigan
Dearborn, MI
houyidu@umich.edu

**Zheng Song**
CIS Department
University of Michigan
Dearborn, MI
zhesong@umich.edu

**Jin Lu**
School of Computing
University of Georgia
Athens, GA
jin.lu@uga.edu

## Abstract

This paper explores the challenges of implementing Federated Learning (FL) in practical scenarios featuring isolated nodes with data heterogeneity, which can only be connected to the server through wireless links in an infrastructure-less environment. To overcome these challenges, we propose a novel mobilizing personalized FL approach, which aims to facilitate mobility and resilience. Specifically, we develop a novel optimization algorithm called Random Walk Stochastic Alternating Direction Method of Multipliers (RWSADMM). RWSADMM capitalizes on the server's random movement toward clients and formulates local proximity among their adjacent clients based on hard inequality constraints rather than requiring consensus updates or introducing bias via regularization methods. To mitigate the computational burden on the clients, an efficient stochastic solver of the approximated optimization problem is designed in RWSADMM, which provably converges to the stationary point almost surely in expectation. Our theoretical and empirical results demonstrate the provable fast convergence and substantial accuracy improvements achieved by RWSADMM compared to baseline methods, along with its benefits of reduced communication costs and enhanced scalability.

## 1 Introduction

Federated Learning (FL) [1, 2, 3, 4] is a distributed machine learning paradigm that enables clients to learn a shared model without sharing their private data. Unlike traditional machine learning approaches that rely on central servers for model training, FL allows clients to collaborate and train the model in a distributed manner, overcoming privacy issues related to passing data to a central server. Despite its advancements, real-world applications in environments with insufficient network support continue to face challenges. a) Maintaining consistent and reliable connections between the central server and clients becomes exceedingly challenging in environments lacking network infrastructures, e.g., natural disasters or military warzones. While intermittent connectivity may be available through satellite networks, the instability and limited capacity of such networks prevent the transmission of large data volumes, making it difficult to collect model updates from soldiers or first responders. b) The non-IID (non-independent and identically distributed) nature of clients' data, characterized by heterogeneity across the network, can hinder the generalization of the global model

37th Conference on Neural Information Processing Systems (NeurIPS 2023), New Orleans, USA.

for each client. Addressing these challenges is crucial for practical FL in such environments. In this paper, we propose RWSADMM, a novel FL scheme that uses Random Walk (RW) algorithm to enable server mobility among client clients. These dynamic approach benefits scenarios with limited internet connectivity, where clients form clusters using local short-range transmission devices.

For example, in various contexts such as robotics, emergency response, or military operations, consider a scenario where individuals or entities are equipped with integrated visual augmentation systems (IVAS) [5]. To facilitate the collection of model updates from these entities, a mobile unit equipped with a powerful computer navigates the environment [6], communicating with locations through a network, which might be satellite-based or another suitable alternative. Upon reaching an entity, the mobile unit employs short-range communication technologies such as WiFi direct, Zigbee [7], or Bluetooth to establish connections with nearby IVAS devices. Through these connections, the unit collects model updates and distributes new models as necessary. A graph-based representation is utilized to determine the order of interactions, where entities are depicted as nodes, and the edge between an entity and its neighbor indicates that the neighbor is within the communication range of the unit that reaches the entity. This graph assists the mobile unit in making informed decisions about the order in which it engages with the entities.

Various applicable examples of such constrained network situations span across different domains, including ad hoc wireless learning [8, 9, 10], wildlife tracking [11, 12, 13], Internet of Underwater Things (IoUT) [14, 15], natural disaster management [16, 17], military operations [18], or fostering a digital democracy [19, 20] which assists in overcoming restrictions imposed by regimes that prohibit internet access to civilians.

Specifically, to address challenge a), we propose an algorithmic framework called RWSADMM (Fig. 1), short for Random Walk Alternating Directional method of Multipliers, which considers a dynamic reachability graph among distributed clients using a movable vehicle as the central server. Clients are represented as nodes in the graph, with edges denoting neighborhood connections. Personal devices, referred to as local clients, establish dynamic connectivity with the server when the server is nearby. The server connects with a selected client and its neighbors while moving between locations using a non-homogeneous RW algorithm for probabilistic navigation. In each computation round, the vehicle broadcasts and gathers local model updates from residing clients, who rely on short-range communication to interact with the moving server, when it's within the communication range. The received updates are aggregated and used to update the global model iteratively.

To tackle the second challenge (b) arising from the heterogeneity of data distribution among clients, RWSADMM incorporates model personalization through local proximity among adjacent clients using hard inequality constraints, as opposed to unconstrained optimization with regularization techniques that may induce model bias. By formulating the problem with these constraints, RWSADMM reduces the computational complexity for clients, effectively mitigating the limitations of local computational power. This is achieved by designing the solver to the stochastic approximation of the minimization subproblem within the typical ADMM algorithm.

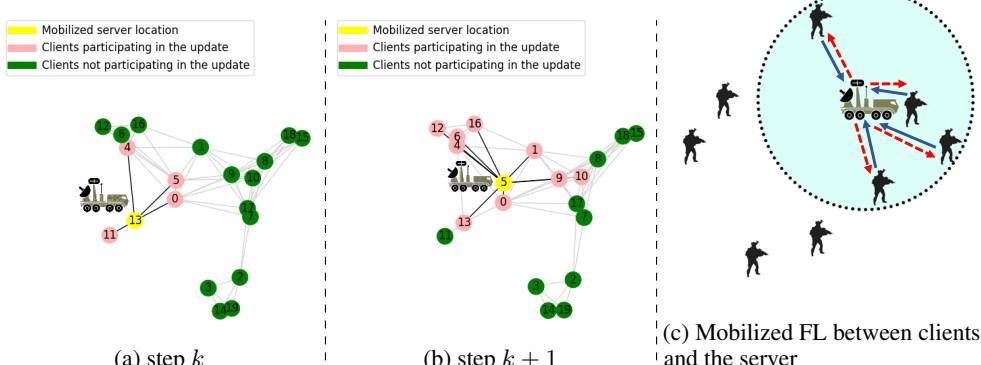

(a) step $k$     (b) step $k+1$     (c) Mobilized FL between clients and the server

Figure 1: This illustration showcases the training process using the RWSADMM algorithm. A vehicle, serving as the mobile server, navigates between different clients using a random walk strategy. (a) In step $k$, the server moves to client 13, covering the clients in $\mathcal{N}(13)$ for FL model training and completing the aggregation step. (b) In step $k+1$, client 5 is selected, and the vehicle moves to client 5. The training and aggregation steps occur within the zone encompassing $\mathcal{N}(5)$.

Our research makes three main contributions. Firstly, our proposed RWSADMM algorithm is **the first attempt to enable mobilizing FL with efficient communication and computation in an infrastructure-less setting**. The RWSADMM framework involves a server dynamically moving between different regions of clients and receiving updates from one or a few clients residing in the selected zone, which reduces communication costs and enhances system flexibility. Secondly, to address the issue of data heterogeneity among clients, RWSADMM **formulates local proximity among adjacent clients based on hard inequality constraints, avoiding the introduction of model bias via consensus updates**. This approach provides an alternative realization of personalization, which is crucial when dealing with highly heterogeneous data distributions. Thirdly, to mitigate the computational burden on the clients, an efficient stochastic solver of the subproblem is designed in RWSADMM, which **provably converges to the stationary point almost surely in expectation** under mild conditions independent of the data distributions. Our theoretical and empirical results demonstrate the superiority of our proposed algorithm over state-of-the-art personalized Federated Learning (FL) algorithms, providing empirical evidence for the effectiveness of our approach.

## 2 Related Work

This paper is relevant to two distinct research areas, which are reviewed in two separate sections: FL frameworks that tackle data heterogeneity and ADMM-based FL frameworks.

**FL with data heterogeneity** FL was initially introduced as FedAvg, a client-server-based framework that didn't allow clients to personalize the global model to their local data [1]. This led to poor convergence due to local data heterogeneity, negatively impacting the global FL model's performance on individual clients. Recent works proposed a two-stage approach to personalize the global model. In the first stage, the FL global model is trained similarly to FedAvg. However, the second stage is included to personalize the global model for each local client through additional training on their local data. [21] demonstrated that FedAvg is equivalent to Reptile, a new meta-learning algorithm, when each client collects the same amount of local data. To learn a global model that performs well for most participating clients, [22] proposed an improved version, Per-FedAvg. This new variant aims to learn a good initial global model that can adapt quickly to local heterogeneous data. An extension of Per-FedAvg, called pFedMe [23], introduced an $\ell_2$-norm regularization term to balance the agreement between local and global models and the empirical loss. [24] proposed Ditto, a multi-task learning-based FL framework that provides personalization while promoting fairness and robustness to byzantine attacks. Ditto uses a regularization term to encourage personalized local models to be close to the optimal global model. [25] proposed to interpolate local and global models to train local models while also contributing to the global model. However, scaling these approaches can be challenging due to high communication costs, reliance on strong assumptions about network connectivity, or the requirement to compute second-order gradients. Additionally, there is a potential for enhancing the algorithms' overall performance.

**ADMM-based FL** The Alternating Direction Method of Multipliers (ADMM) is a widely recognized algorithm that effectively tackles optimization problems across multiple domains. In recent studies, ADMM has been successfully employed in distributed learning, as demonstrated by several works [26, 27, 28, 29, 30, 31, 32, 33]. In Federated Learning (FL) context, researchers have proposed various methods to address specific challenges. For handling falsified data in Byzantine settings, [34] introduced a robust ADMM-based approach. To mitigate local computational burdens in FL, [35] developed an inexact ADMM-based algorithm suitable for edge learning configurations. FL itself enables local training without the need to share personal data between clients and the server. Despite the advantages of FL, there is still a concern regarding clients' privacy. Analyzing the parameter differences in the trained models uploaded by each client can compromise their privacy. To tackle this issue, [36] proposed an inexact ADMM-based federated learning algorithm that incorporates differential privacy (DP) techniques [37]. By leveraging DP, the algorithm enhances privacy protection during the FL process. These ADMM-based frameworks also have high communication costs, ranging from $O(n)$ to $O(n^2)$ per iteration, depending on the network's density with $n$ clients. [38] introduced a Proximal Primal-Dual Algorithm (Prox-PDA) to enable network nodes to compute the set of first-order stationary solutions collectively. Moreover, these algorithms do not account for data heterogeneity in their framework designs, leading to performance deterioration in such scenarios. The most similar algorithm to RWSADMM is called Walkman [39]. Walkman is an ADMM-based framework utilizing the random walk technique for distributed optimization. In

Walkman, the communication and computation costs are reduced by activating only one agent at each step. Compared to other ADMM-based approaches, including Walkman, RWSADMM has several distinctive features. RWSADMM leverages stochastic approximation to reduce computation costs per iteration and enforces hard inequality constraints instead of consensus to manage heterogeneous data, resulting in increased robustness. Additionally, RWSADMM considers the dynamic graph, allowing it to adapt to changing network conditions and potentially improve communication efficiency. RWSADMM also incorporates a hard constraint parameter $\epsilon$ to promote local proximity among clients instead of using a regularization term as Walkman does to promote client consensus. This approach better balances personalization and global optimization. Finally, while Walkman is fully distributed without server involvement, RWSADMM is a server-based approach in which the server aggregates information from a small group of clients in each computation round.

## 3 Random Walk Stochastic ADMM (RWSADMM)

Before delving into the specifics of the proposed algorithm, we present the key notation used throughout this research. $\mathbf{x} \in \mathbb{R}^d$ represents a vector with length $d$ and $\mathbf{e}$ is defined as a vector with entries equal to 1 and $\mathbf{X} \in \mathbb{R}^{l \times d}$ depicts a matrix with $l$ rows and $d$ columns. $[\mathbf{x}]_i$ represents the $i$th element of vector $\mathbf{x}$ and $[\mathbf{X}]_{ij}$ is the $(i,j)$th element of matrix $\mathbf{X}$. $[\mathbf{X}]_i$, $[\mathbf{X}]^j$ represent the $i$th row and $j$th column of matrix $\mathbf{X}$, respectively. $(\nabla f(\mathbf{x}))_j$ is used to denote the $j$th entry of the gradient of $f(\mathbf{x})$. The inner product of $A$ and $B$ is shown as $\langle A, B \rangle$. $\mathbb{E}_t[.]$ indicates the expectation given the past $\xi_1, \ldots, \xi_{t-1}$. $\odot$ represents the Hadamard product/element-wise product and $\otimes$ represents the Kronecker product between two matrices. Finally, Norm p of vector $\mathbf{x}$ is denoted as $||\mathbf{x}||_p^p = \sum_{i=1}^{d} |x_i|^p$, $\mathbf{x} \in \mathbb{R}^d$ and Frobenius norm of matrix $\mathbf{X}$ is written as $||\mathbf{X}||_F = \sqrt{\sum_{i=1}^{n} \sum_{j=1}^{m} |x_{ij}|^2}$.

Let us first define our Mobilizing FL problem. Mobilizing FL, which involves a mobile server, can be formulated as an optimization problem on a connected graph $\mathcal{G} = (\mathcal{V}, \mathcal{E})$. The graph comprises a set of $n$ clients, represented as $\mathcal{V} = v_1, v_2, \ldots, v_n$, and a set of $m$ edges, denoted as $\mathcal{E}$. The objective is to minimize the average loss function across all clients while adhering to inequality constraints that ensure local proximity among the clients' respective local models. The optimization problem can be formulated mathematically as follows:

$$\min_{\mathbf{x}_{1:n} \in \mathbb{R}^p} \frac{1}{n} \sum_{i=1}^{n} f_i(\mathbf{x}_i) \quad s.t. \quad |\mathbf{x}_i - \mathbf{x}_j| \leq \epsilon_i, \forall i \in \{1, \ldots, n\}, \forall j \in \mathcal{N}(i)/v_i. \tag{1}$$

where $f_i(\mathbf{x}_i)$ represents the local loss function with the model parameter as $\mathbf{x}_i$ for the client $i$, the vertex set $\mathcal{N}(i)$ contains client $i$ and its neighboring clients, and $\epsilon_i$ is the non-consensus relaxation between local neighboring clients to replace model consensus requirement in typical FL. In our proposed FL method, we model the server's movement as a dynamic Markov Chain, introducing a dynamic element to the traditional ADMM-based approach. This work is the first to consider a dynamic mobile server within the ADMM-based FL framework. In RWSADMM, client-server communication occurs only when the server is close to a client. The sequence of client indices that are updated, denoted as $i_k$, evolves based on a non-homogeneous Markov Chain with a state space of $1, \ldots, n$ [40]. To describe the transition dynamics of the Markov Chain, we employ the non-homogeneous Markovian transition matrix $\mathbf{P}(k)$, which represents the probabilities of transitioning between clients at time $k$. Specifically, the conditional probability of selecting client $j$ as the next client, given that client $i$ is the current client, is defined as:

$$[\mathbf{P}(k)]_{i,j} = Pr\{i_{k+1} = j | i_k = i\} \in [0, 1] \tag{2}$$

Additionally, it is assumed that the server determines the probability of all possible locations for its next destination based on the transition matrix $\mathbf{P}(k)$ at time $k$. This provides a probabilistic approach to server navigation, allowing it to move around the network more effectively. To guarantee convergence, RWSADMM depends on the frequency of revisiting each agent. This quality is described by the *mixing time* of the algorithm. An assumption for the mixing time is as follows:

**Assumption 3.1.** The random walk $(i_k)_{k \geq 0}, v_{i_k} \in \mathcal{V}$ forms an irreducible and aperiodic (ergodic) Markov Chain with transition probability matrix of $\mathbf{P}(k) \in \mathbb{R}^{n \times n}$ defined in Eq. (2) and stationary distribution $\boldsymbol{\pi}$ satisfying $\lim_{k \to \infty} \boldsymbol{\pi}^T \mathbf{P}(k) = \boldsymbol{\pi}^T$. The mixing time (for a given $\delta > 0$) is defined as the smallest integer $\tau(\delta)$ such that $\forall i \in V$,

$$\left\| [\mathbf{P}(k)^{\tau(\delta)}]_i - \boldsymbol{\pi}^{\mathbf{T}} \right\| \leq \delta \boldsymbol{\pi}_* \tag{3}$$

where $\boldsymbol{\pi}_* := \min_{i \in \mathcal{V}} \boldsymbol{\pi}_i$. This inequality states the fact that regardless of the current state $i$ and time $k$, the probability of visiting each state $j$ after $\tau(\delta)$ steps is $(\delta \boldsymbol{\pi}_*)$-close to $\boldsymbol{\pi}_j$, that is, $\forall i, j \in \mathcal{V}$,

$$\left\| [\mathbf{P}(k)^{\tau(\delta)}]_{ij} - \boldsymbol{\pi}_j \right\| \leq \delta \boldsymbol{\pi}_* \tag{4}$$

Eq. (4) is used to prove the sufficient descent of a Lyapunov function $L_\beta$ in Section 3.1. Let's also define

$$\mathbf{P}_{max} = \lim_{k=+\infty} \{ \mathbf{P} | [\mathbf{P}]_{ij} = \max_k [\mathbf{P}(k)]_{ij} \}, \tag{5}$$

from which one can further obtain $\|\mathbf{P}(k)\| \leq \|\mathbf{P}_{max}\|$ for all $k$. Namely, the matrix $P_{max}$ is computed as the element-wise maximum matrix among all the matrices $P(k)$, for $k = 0, \ldots, \infty$. Therefore, the mixing time requirement in Eq. (3) is guaranteed to hold for

$$\tau(\delta) = \lceil \frac{1}{1 - \sigma(\mathbf{P})} \ln \frac{\sqrt{2}}{\delta \pi_*} \rceil \overset{(a)}{\leq} \lceil \frac{1}{1 - \sigma(\mathbf{P}_{max})} \ln \frac{\sqrt{2}}{\delta \pi_*} \rceil \tag{6}$$

where $\sigma(\mathbf{P}) := \sup\{\|f^T \mathbf{P}\|/\|f\| : f^T \mathbf{1} = 0, f \in \mathbb{R}^n\}$. Using Eq. (5), we have $\forall \mathbf{P}, \ \sigma(\mathbf{P}) \leq \sigma(\mathbf{P})_{max}$ and the inequality $(a)$ can be inferred.

## 3.1 Algorithm

In this section, we derive RWSADMM by integrating random walk and stochastic inexact approximation techniques into ADMM. Considering $\mathbf{X} := row(\mathbf{x}_1, \mathbf{x}_2, \ldots, \mathbf{x}_n) \in \mathbb{R}^{p \times n}$, $F(\mathbf{X}) := \sum_{i=1}^n f_i(\mathbf{x}_i)$, where the operation $row(.)$ refers to row-wise stacking of vectors $\mathbf{x}_i$'s. The mobilizing FL problem (1) can be expressed as:

$$\min_{\mathbf{y}_{1:n}, \mathbf{X}} \frac{1}{n} F(\mathbf{X}) \quad \text{s.t.} \ \left| \mathbf{1} \otimes \mathbf{y}_i - \mathbf{X}_{\mathcal{N}(i)} \right| \leq \mathbf{1} \otimes \boldsymbol{\epsilon}_i/2, \forall i = 1, \ldots, n \tag{7}$$

where $\mathbf{1} = [1 \ 1 \ldots 1] \in \mathbb{R}^{n_i}$, $n_i$ denotes the volume of the vertex set $\mathcal{N}(i)$. The constraint implies that $|\mathbf{x}_i - \mathbf{x}_j| \leq \boldsymbol{\epsilon}_i$, $\forall i = 1 \ldots n$ and $\forall j \in \mathcal{N}(i)$ through the triangle inequality. $\mathbf{y}_i$ stored on the server is necessarily introduced as a local proximity of $\mathcal{N}(i)$. We can obtain the augmented Lagrangian for problem (7)

$$L_\beta(\mathbf{y}_{1:n}, \mathbf{X}, \mathbf{Z}_{1:n}) = \frac{1}{n} \Big[ F(\mathbf{X}) + \sum_{i=1}^n \langle \mathbf{Z}_i, \left| \mathbf{1} \otimes \mathbf{y}_i - \mathbf{X}_{\mathcal{N}(i)} \right| - \boldsymbol{\varepsilon}_i \rangle + \frac{\beta}{2} \sum_{i=1}^n \| \left| \mathbf{1} \otimes \mathbf{y}_i - \mathbf{X}_{\mathcal{N}(i)} \right| - \boldsymbol{\varepsilon}_i \|_F^2 \Big] \tag{8}$$

where $\boldsymbol{\varepsilon}_i = \boldsymbol{\epsilon}_i/2$ and $\mathbf{Z}_i \in \mathbb{R}^{n_i p}$ are the dual variable and $\beta > 0$ is the barrier parameter. The RWSADMM algorithm minimizes the augmented Lagrangian $L_\beta(\mathbf{y}_{1:n}, \mathbf{X}, \mathbf{Z}_{1:n})$ in an iterative manner. At each iteration $k$, only a subset of clients covered by the mobilized server, the clients in $\mathcal{N}(i_k)$, participate in the federated update. The following updates are performed:

$$\mathbf{x}_{i_k} = \arg\min_{\mathbf{x}_{i_k}} L_\beta(\mathbf{y}'_{i_k}, \mathbf{x}_{i_k}, \mathbf{z}'_{i_k}), \quad \mathbf{y}_{i_k} = \arg\min_{\mathbf{y}_{i_k}} L_\beta(\mathbf{y}_{i_k}, \mathbf{X}_{\mathcal{N}(i_k)}, \mathbf{Z}'_{\mathcal{N}(i_k)}),$$

where $\mathbf{y}'_{i_k}$, $\mathbf{x}_{i_k}$, and $\mathbf{z}'_{i_k}$ denote the groups of variables of the local parameters stored by client $i_k$ at the $(k-1)th$ update. After solving these subproblems, we update the multiplier $\mathbf{z}_{i_k}$ as follows:

$$\mathbf{z}_{i_k} = \mathbf{z}'_{i_k} + \beta(\left| \mathbf{y}_{i_k} - \mathbf{x}_{i_k} \right| - \boldsymbol{\varepsilon}_i),$$

Next, we derive the solver of each subproblem. The three steps are noted as Updating $\mathbf{x}_{i_k}$, Updating $\mathbf{y}_{i_k}$, and Updating $\mathbf{z}_{i_k}$.

Updating $\mathbf{x}_{i_k}$: $\quad \min_{\mathbf{x}_{i_k}} \Big[ f_{i_k}(\mathbf{x}_{i_k}) + \langle \mathbf{z}'_{i_k}, \left| \mathbf{y}'_{i_k} - \mathbf{x}_{i_k} \right| - \boldsymbol{\varepsilon}_{i_k} \rangle + \frac{\beta}{2} \| \left| \mathbf{y}'_{i_k} - \mathbf{x}_{i_k} \right| - \boldsymbol{\varepsilon}_{i_k} \|_2^2 \Big]$ $\tag{9}$

The Problem (9) can be solved iteratively, consuming significant computational resources for the local clients. Furthermore, the computational complexity increases as the local dataset grows, as is often true in real-world applications. By utilizing the stochasticity and first-order subgradient expansion, we arrive at a more computationally efficient approximation of the original problem in Eq. (10).

$$\min_{\mathbf{x}_{i_k}} \Big[ g_{i_k}(\mathbf{x}'_{i_k}, \boldsymbol{\xi}_{i_k})(\mathbf{x}_{i_k} - \mathbf{x}'_{i_k}) + \langle \mathbf{z}'_{i_k}, \left| \mathbf{y}'_{i_k} - \mathbf{x}_{i_k} \right| - \boldsymbol{\varepsilon}_{i_k} \rangle + \frac{\beta}{2} \| \left| \mathbf{y}'_{i_k} - \mathbf{x}_{i_k} \right| - \boldsymbol{\varepsilon}_{i_k} \|_2^2 \Big] \tag{10}$$

In Eq. (10), $\boldsymbol{\xi}_{i_k}$ denotes one or a few samples randomly selected by client $i_k$ from its feature set and their ground truth labels in pairs at the $k$-th iteration. The function $g_{i_k}(\mathbf{x}'_{i_k}, \boldsymbol{\xi}_{i_k})$ is defined as

the stochastic gradient of $f_{i_k}(\mathbf{x}'_{i_k})$ at $\mathbf{x}'_{i_k}$. The stochastic approximation can tremendously reduce memory consumption and save computational costs in each iteration. By setting the subgradient of the objective function in Eq. (10) to zero, we can derive the closed-form solution in Eq. (11).

$$\mathbf{x}_{i_k} = \mathbf{y}'_{i_k} + \frac{1}{\beta}\mathbf{z}'_{i_k} \odot sgn(\mathbf{t}') - \frac{1}{\beta}sgn(\mathbf{t}') \odot \left(\varepsilon_i + g_{i_k}(\mathbf{x}'_{i_k}, \xi_{i_k})\right) = \mathbf{y}'_{i_k} + \frac{1}{\beta}sgn(\mathbf{t}') \odot (\mathbf{z}'_{i_k} - \varepsilon_i - g_{i_k}(\mathbf{x}'_{i_k}, \xi_{i_k})) \quad (11)$$

where the signum function $sgn(\cdot)$ extracts the signs of a vector and $\mathbf{t}'_{i_k} = \mathbf{y}'_{i_k} - \mathbf{x}'_{i_k}$.

Updating $\mathbf{y}_{i_k}$: We solve the following problem

$$\min_{\mathbf{y}_{i_k}} \langle \mathbf{Z}_{\mathcal{N}(i_k)}, |\mathbf{1} \otimes \mathbf{y}_{i_k} - \mathbf{X}_{\mathcal{N}(i_k)}| - \mathbf{1} \otimes \varepsilon_{i_k} \rangle + \frac{\beta}{2} \||\mathbf{1} \otimes \mathbf{y}_{i_k} - \mathbf{X}_{\mathcal{N}(i_k)}| - \mathbf{1} \otimes \varepsilon_{i_k}\|_F^2 \quad (12)$$

one can readily derive a closed-form solution for the problem (12) as:

$$\mathbf{y}_{i_k} = \frac{1}{n_{i_k}} \sum_{j \in \mathcal{N}_{i_k}} \left[\mathbf{x}_{i_k} - (\frac{\mathbf{z}_{i_k}}{\beta} + \varepsilon_{i_k}) \odot sgn(\mathbf{t}_{i_k})\right] \quad (13)$$

where $\mathbf{t}_{i_k} = \mathbf{y}'_{i_k} - \mathbf{x}_{i_k}$ is similar to that of Eq. (11) except the updated $\mathbf{x}$. Specifically, via mathematical induction, we can attain the new updated form of $\mathbf{y}_{i_k}$ below, which can also reduce the communication cost from $O(n)$ to $O(1)$:

$$\mathbf{y}_{i_k} = \mathbf{y}'_{i_k} + \frac{1}{n_{i_k}}\left[\mathbf{x}_{i_k} - (\frac{\mathbf{z}_{i_k}}{\beta} + \varepsilon_{i_k}) \odot sgn(\mathbf{t}_{i_k})\right] - \left[\mathbf{x}'_{i_k} - (\frac{\mathbf{z}'_{i_k}}{\beta} + \varepsilon_{i_k}) \odot sgn(\mathbf{t}'_{i_k})\right] \quad (14)$$

Updating $\mathbf{z}_{i_k}$: The Lagrangian multiplier $\mathbf{z}_{i_k}$ can be updated strictly following the standard ADMM scheme below:

$$\mathbf{z}_{i_k} = \mathbf{z}'_{i_k} + \kappa\beta\left[\mathbf{x}_{i_k} - \mathbf{y}'_{i_k} - \varepsilon_{i_k}\right] \quad (15)$$

The $\kappa$ coefficient used in Eq. (15) is decayed in each process step to achieve better convergence.

Please refer to Appendix A for the entire RWSADMM algorithm framework.

## 4   Theoretical Analysis

In this section, we present the theoretical convergence guarantee of RWSADMM. To ensure its convergence, certain common assumptions are made regarding the properties of the loss functions. The assumptions are as follows:

**Assumption 4.1.** The objective function $f(\mathbf{x})$ is bounded from below and coercive over $\mathbb{R}^p$, that is, for any sequence $\{\mathbf{x}^k\}_{k \geq 0} \subset \mathbb{R}^p$,

$$\text{if } \|\mathbf{x}^k\| \xrightarrow{k \to \infty} \infty \Rightarrow \frac{1}{n}\sum_{i=1}^{n} f_i(\mathbf{x}) \to \infty \quad (16)$$

**Assumption 4.2.** The objective function $f_i(\mathbf{x})$'s are L-smooth, that is, $f_i$ are differentiable, and its gradients are L-Lipschitz, that is, $\forall \mathbf{u}, \mathbf{v} \in \mathbb{R}^p$ [39],

$$\|\nabla f_i(\mathbf{u}) - \nabla f_i(\mathbf{v})\| \leq L\|\mathbf{u} - \mathbf{v}\|, \quad \forall i = 1, \ldots, n \quad (17)$$

Remark: In consequence it also holds that $\forall \mathbf{u}, \mathbf{v} \in \mathbb{R}^p$

$$f_i(\mathbf{u}) - f_i(\mathbf{v}) \leq \nabla f_i(\mathbf{v})^T(\mathbf{u} - \mathbf{v}) + \frac{L}{2}\|\mathbf{u} - \mathbf{v}\|^2, \quad \forall i = 1, \ldots, n. \quad (18)$$

**Assumption 4.3.** The objective function $f$ is M-Lipschitz, that is, $\forall \mathbf{u}, \mathbf{v} \in \mathbb{R}^p$ [41],

$$|f(\mathbf{u}) - f(\mathbf{v})| \leq M\|\mathbf{u} - \mathbf{v}\| \quad (19)$$

**Assumption 4.4.** The first-order stochastic gradient is sampled, which returns a noisy but unbiased estimate of the gradient of $f$ at any point $\mathbf{x} \in \mathbb{R}^p$, that is, $\forall \mathbf{x} \in \mathbb{R}^p$,

$$\mathbb{E}_\xi[g(\mathbf{x}, \xi)] = \nabla f(\mathbf{x}) \quad (20)$$

Remark: Substituting Eq. (20) into Eq. (17), one can obtain that for $i = 1, \ldots, n$, we have

$$\|\mathbb{E}_\xi[g(\mathbf{u}, \xi)] - \mathbb{E}_\xi[g(\mathbf{v}, \xi)]\| \leq L\|\mathbf{u} - \mathbf{v}\| \quad (21)$$

Substituting Eq. (20) into Eq. (18), for $i = 1, \ldots, n$, we can obtain

$$f_i(\mathbf{u}) - f_i(\mathbf{v}) \leq \mathbb{E}_\xi[g(\mathbf{v}, \xi)]^T(\mathbf{u} - \mathbf{v}) + \frac{L}{2}\|\mathbf{u} - \mathbf{v}\|^2, \quad (22)$$

**Assumption 4.5.** The noise variance of the stochastic gradient is bounded as:

$$\mathbb{E}_\xi(\|\nabla f(\mathbf{x}) - g(\mathbf{x}, \xi)\|^2) \leq \exp(1), \text{ for all } \mathbf{x}. \tag{23}$$

This condition bounds the expectation of $\|\nabla f(\mathbf{x}_t) - g(\mathbf{x}_t, \xi_t)\|^2$. Using Jensen's inequality, this condition implies a bounded variance [41].

We revisit the related crucial properties of the Markov Chain. The first time that the Markov Chain $(i_k)_{k \geq 0}$ hits agent $i$ is denoted as $T_i := \min\{k : i_k = i\}$, and maximum value of $T$ over all clients is defined as $T := \max\{T_1, \ldots, T_n\}$. For $k > T$, let $\tau(k, i)$ denote the iteration of the last visit to agent $i$ before $k$, mathematically we have

$$\tau(k, i) = \max\{k' : i_{k'} = i, k' < k\}. \tag{24}$$

To prove the convergence of our proposed algorithm, two Lyapunov functions defined for RWSADMM are required to be investigated:

$$L_\beta^k := L_\beta(\mathbf{y}^k, \mathbf{X}^k; \mathbf{Z}^k), \quad M_\beta^k := L_\beta^k + \frac{L^2}{n} \sum_{i=1}^n \left\| \mathbf{y}_i^{\tau(k,i)+1} - \mathbf{y}_i^{\tau(k,i)} \right\|^2 \tag{25}$$

where $L_\beta(\mathbf{y}^k, \mathbf{X}^k; \mathbf{Z}^k)$ is defined in Eq. (8). The $M_\beta^k$ is utilized in the convergence analysis. To guarantee the convergence of our algorithm, first, we refer to the asymptotic analysis of the nonhomogeneous Markov chain presented in [42]. Define $\mathbf{\Phi}(k, l)$ with $k \geq l$ as the product of the transition probability matrices for the Markov chain from time $l$ to $k$, i.e., $\mathbf{\Phi}(k, l) = \mathbf{P}(k) \ldots \mathbf{P}(l)$ with $k \geq l$. Then we have the following convergence result:

**Lemma 4.6.** Consider

1. $\forall s, \lim_{k \to \infty} \mathbf{\Phi}(k, l) = \frac{1}{n} \mathbf{e} \mathbf{e}^T$.
2. The convergence of $\mathbf{\Phi}$ is geometric and the rate of convergence considering $\forall k, l$, with $k \geq l \geq 0$, is given by

$$\left| [\mathbf{\Phi}(k, l)]_{i,j} - \frac{1}{n} \right| \leq \left(1 - \frac{\eta}{4n^2}\right)^{\lceil \frac{k-l+1}{Q} \rceil - 2} \tag{26}$$

Using Lemma 4.6, the convergence analysis of the algorithm is as follows.

**Lemma 4.7.** Under Assumptions 4.1 and 4.2, if $\beta > 2L^2 + L + 2$, $(M_\beta^k)_{k \geq 0}$ is lower bounded and convergent, the iterates $(\mathbf{y}^k, \mathbf{X}^k, \mathbf{Z}^k)_{k \geq 0}$ generated by RWSADMM is bounded.

The proof sketch and the detailed convergence proof are presented in Appendix B. Using Lemma 4.7 and B.6, we can present the convergence of RWSADMM in Theorem 4.8.

**Theorem 4.8.** *Let Assumption 4.5 hold. For $\beta > 2L^2 + L + 2$, it holds that any limit point $(\mathbf{y}^*, \mathbf{X}^*, \mathbf{Z}^*)$ of the sequence $(\mathbf{y}^k, \mathbf{X}^k, \mathbf{Z}^k)$ generated by RWSADMM satisfies $\mathbf{y}^* = \mathbf{x_i}^*$, $i = 1, \ldots, n$ where $\mathbf{y}^*$ is a stationary point of Eq. (7), with probability 1, that is,*

$$Pr\left(0 \in \frac{1}{n} \sum_{i=1}^n \nabla f_i(\mathbf{y}^*)\right) = 1 \tag{27}$$

*If the objective function of Eq. (7) is convex, then $\mathbf{y}^*$ is a minimizer.*

Next, Theorem 4.9 further presents that the algorithm converges sublinearly. This is comparable to the convergence rate of other FL methods [43, 44, 24, 25], but the existing methods didn't consider the dynamic graph and infrastructure-less environment. The detailed proof is offered in Appendix C.

**Theorem 4.9.** *Under Assumptions (3.1), (4.1), and (4.2), with given $\beta$ in Lemma 4.7, and local variables initiated as $\nabla f_i(\mathbf{x}_i^0) = \beta \mathbf{x}_i^0 = \mathbf{z}_i^0, \forall i \in \{1, \ldots, n\}$, there exists a sequence $\{g^k\}_{k \geq 0}$ with $\{g^k\} \in \partial L_\beta^{k+1}$ satisfying*

$$\min_{k \leq K} \mathbb{E} \|g^k\|^2 \leq \frac{C}{K}(L_\beta^0 - \underline{f}), \quad \forall K > \tau(\delta) + 2 \tag{28}$$

*where $C$ is a constant depending on $\beta$, $L$, and $\gamma$, $n$, and $\tau(\delta)$.*

**Communication Complexity** Using Theorem 4.9, the communication complexity of RWSADMM for nonconvex nonsmooth problems is as follows. To achieve ergodic gradient deviation $E_t := \min_{k \leq K} \mathbb{E} \|g^k\|^2 \leq \omega$ for any $K > \tau(\delta) + 2$, it is sufficient to have

$$\frac{C}{K}(L_\beta^0 - f) \le \omega \xrightarrow{(a)} K \sim O\left(\frac{1}{\omega} \cdot \frac{\tau(\delta)^2 + 1}{(1-\delta)n\pi_*}\right) \tag{29}$$

(a) is achieved by taking $L_\beta^0$ and $f$ as constants and independent of $n$ and the network structure. Using the $\tau(\delta)$ definition from (6), by setting $\delta = 1/2$ and assuming the reversible Markov chain with $P(k)^T = P(k)$, the communication complexity is

$$O\left(\frac{1}{\omega} \cdot \frac{ln^2 n}{(1-\lambda_2(\mathbf{P}(k)))^2}\right) \tag{30}$$

**Communication Comparison** Among the baseline frameworks, Per-FedAvg [22] and APFL [25] have addressed the communication complexity of their respective frameworks. By assuming that Assumption 3.1 holds and utilizing Eq. (30), we can determine the communication complexity of RWSADMM as $O(\omega^{-1})$ for $K$ iterations. In comparison, Per-FedAvg exhibits a higher communication complexity of $O(\omega^{-3/2})$. In the case of APFL, all clients are assumed to be used in each computation round to ensure convergence in nonconvex settings. The communication complexity of APFL is determined as $O(n^{3/4}\omega^{-3/4})$, where $n$ represents the total number of clients. Consequently, when $n$ is large, APFL exhibits a significantly higher communication rate than RWSADMM. Overall, the communication complexity analysis suggests that RWSADMM offers superior scalability and communication efficiency compared to existing methods.

## 5 Experimental Results

**Setup** We evaluate the performance of RWSADMM using heterogeneous data distributions. All the experiments are conducted on a workstation with Threadripper Pro 5955WX, 64GB DDR4 RAM, and NVidia 4090 GPU. All frameworks are performed on standard FL benchmark datasets (MNIST [45], Synthetic [23], and CIFAR10 [46]) with 10-class labels and convex and non-convex models. Multinomial logistic regression (MLR), multilayer perceptron network (MLP), and convolutional neural network (CNN) models are utilized for strongly convex and two non-convex settings, respectively. We create a moderately dynamic connected graph of randomly placed nodes where each node has at least 5 neighboring nodes at $k$-th update. We set the probability transition matrix $\mathbf{P}(k)$ as $[\mathbf{P}(k)]ij = 1/deg(i_k)$ and set up the experiments for $N = 20$ clients with a regeneration frequency of 10 steps for the dynamic graph. The data is split among clients using a pathological non-IID setting. The data on each client contains a portion of labels (two out of ten labels), and the allocated data size for each client is variable. For the Synthetic data, we use the same data generative procedures of [23] with 60 features and 100 clients. All local datasets are split randomly with 75% and 25% for training and testing, respectively. The models' details, the rationale behind graph construction, and hyperparameter tuning for $\beta$, $\kappa$, and selected $\varepsilon$ value are further described in Appendix D.

**Performance Comparison** The performance of RWSADMM is compared with FedAvg [1] as a benchmark and several state-of-the-art personalized FL algorithms such as Per-FedAvg [22], pFedMe [23], APFL [25], and Ditto [24]. The test accuracy and training loss for the MNIST dataset is depicted in Fig. 2. (Synthetic and CIFAR10 figures are presented in Appendix D). Test accuracy and time cost for all the datasets are reported in Table 1.

The test accuracy progress curves of RWSADMM for all the models (2a-2c) have a significantly faster convergence. For the non-convex models (2b), RWSADMM reaches convergence after 200 iterations, while the rest of the algorithms, except Ditto, work toward convergence until 600 iterations. The performance curves are shown for 100 iterations for consistency. When tested on MNIST with MLP, RWSADMM demonstrated comparable performance against pFedMe. In the test on Synthetic data with MLR models, RWSADMM exhibited a significant advantage over the other methods, with an improved margin of 14.95%. Regarding computational efficiency, RWSADMM is slower than FedAvg and Per-FedAvg, but faster than pFedMe. Furthermore, RWSADMM converges in fewer iterations (200 iterations) than pFedMe (600 iterations). RWASDMM is also run for more extensive networks with 50 and 100 nodes as a separate set of experiments. The performance comparison results and diagrams are also in Appendix D.

## 6 Conclusion and Future Work

This study proposes a novel approach called RWSADMM, designed for systems with isolated nodes connected via wireless links to the mobile server without relying on pre-existing communication

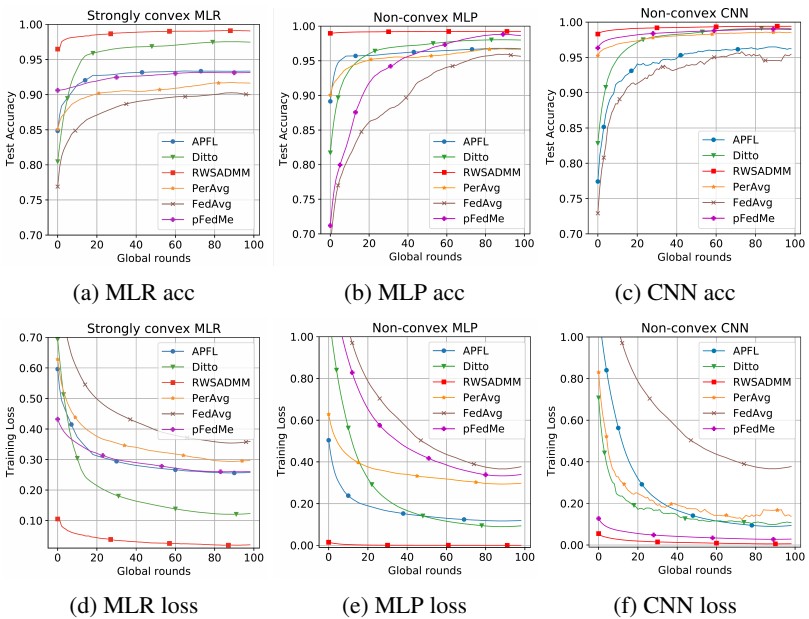

Figure 2: Performance comparison (test accuracy and training loss) of RWSADMM, pFedMe, Per-Avg, FedAvg, APFL, and Ditto for MNIST dataset for the MLR (2a, 2d), MLP (2b, 2e), and CNN (2c, 2f) models. The first 100 iterations are plotted to show the convergence progress better.

| Frameworks | MNIST | | | | | | Synthetic | | | |
|---|---|---|---|---|---|---|---|---|---|---|
| | MLR | | MLP | | CNN | | MLR | | MLP | |
| | acc(%) | t(s) | acc(%) | t(s) | acc(%) | t(s) | acc(%) | t(s) | acc(%) | t(s) |
| FedAvg | $93.96 \pm 0.02$ | 128 | $98.79 \pm 0.03$ | 155 | $97.83 \pm 0.15$ | 2655 | $77.62 \pm 0.11$ | 592 | $83.64 \pm 0.22$ | 680 |
| PerAvg | $94.37 \pm 0.04$ | 154 | $98.90 \pm 0.02$ | 203 | $98.97 \pm 0.08$ | 2432 | $81.49 \pm 0.09$ | 267 | $85.01 \pm 0.10$ | 269 |
| pFedMe | $95.62 \pm 0.04$ | 448 | $\mathbf{99.46 \pm 0.01}$ | 699 | $99.05 \pm 0.06$ | 5541 | $83.20 \pm 0.06$ | 254 | $86.36 \pm 0.15$ | 1413 |
| Ditto | $97.37 \pm 0.02$ | 276 | $97.79 \pm 0.03$ | 423 | $99.20 \pm 0.11$ | 3273 | $86.24 \pm 0.03$ | 72 | $85.26 \pm 0.10$ | 79 |
| APFL | $92.64 \pm 0.03$ | 304 | $97.74 \pm 0.02$ | 533 | $98.58 \pm 0.03$ | 5933 | $83.40 \pm 0.04$ | 95 | $82.52 \pm 0.15$ | 111 |
| RWSADMM (our method) | $\mathbf{98.63 \pm 0.01}$ | 167 | $\mathbf{99.29 \pm 0.02}$ | 295 | $\mathbf{99.52 \pm 0.04}$ | 3857 | $\mathbf{96.44 \pm 0.12}$ | 473 | $\mathbf{97.17 \pm 0.18}$ | 692 |

| Frameworks | CIFAR10 | | | | | |
|---|---|---|---|---|---|---|
| | MLR | | MLP | | CNN | |
| | acc(%) | t(s) | acc(%) | t(s) | acc(%) | t(s) |
| FedAvg | $40.84 \pm 0.01$ | 160 | $41.02 \pm 0.05$ | 69 | $38.65 \pm 0.05$ | 78 |
| PerAvg | $47.43 \pm 0.09$ | 192 | $60.25 \pm 0.07$ | 253 | $83.52 \pm 0.01$ | 800 |
| pFedMe | $67.53 \pm 0.34$ | 515 | $78.12 \pm 0.38$ | 340 | $83.56 \pm 0.05$ | 3480 |
| Ditto | $75.2 \pm 0.01$ | 225 | $81.37 \pm 0.13$ | 259 | $83.86 \pm 0.02$ | 2189 |
| APFL | $75.17 \pm 0.32$ | 50 | $78.00 \pm 0.18$ | 55 | $66.23 \pm 0.03$ | 702 |
| RWSADMM (our method) | $\mathbf{80.72 \pm 0.11}$ | 131 | $\mathbf{84.99 \pm 0.20}$ | 253 | $\mathbf{87.08 \pm 0.03}$ | 3759 |

Table 1: Performance comparisons of FedAvg, Per-FedAvg, pFedMe, Ditto, APFL, and RWSADMM frameworks on MNIST, Synthetic, and CIFAR10 datasets. Three models are utilized for each dataset, and each model's converged accuracy (%) and time consumption (seconds) are reported. Each configuration is executed for ten iterations, and variance is calculated to compute the degree of confidence for test accuracy rates.

infrastructure. The algorithm enables the server to move randomly toward a local client, establishing local proximity among adjacent clients based on hard inequality constraints, addressing the challenge of data heterogeneity. Theoretical and experimental results demonstrate that RWSADMM is fast-converging and communication-efficient, surpassing current state-of-the-art FL frameworks. This study primarily focuses on the methodological framework for RWSADMM. Future research directions should explore essential techniques such as incorporating differential privacy techniques and examining scalability in more extensive network and dataset scenarios. Further investigation is needed to assess the implementation in physical networks and evaluate the effect of communication delays in the real world.

# 7 Acknowledgement

This research was partially supported by the NSF 2122309 and NSF 2104337.

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
