## A    RWSADMM: Algorithm

The RWSADMM scheme is as presented in Algorithm 1. Client $i_k$ is selected by Random Walk via $P(k)$, and $\mathbf{y}'_{i_k}$ is the token from the previous update. Note that we only use one client in each derivation iteration. Still, it is straightforward to generalize the algorithm to have multiple active clients in $\mathcal{S}(i_k) \subset \mathcal{N}(i_k)$ simultaneously to stabilize the computation better as follows:

$$\mathbf{y}_{i_k} = \mathbf{y}'_{i_k} + \frac{1}{n_{i_k} S_{i_k}} \sum_{i_k \in \mathcal{S}(i_k)} \left[ \mathbf{x}_{i_k} - (\frac{\mathbf{z}_{i_k}}{\beta} + \boldsymbol{\varepsilon}_{i_k}) \odot sgn(\mathbf{t}_{i_k}) \right] - \sum_{i_k \in \mathcal{S}(i_k)} \left[ \mathbf{x}'_{i_k} - (\frac{\mathbf{z}'_{i_k}}{\beta} + \boldsymbol{\varepsilon}_{i_k}) \odot sgn(\mathbf{t}'_{i_k}) \right] \quad (31)$$

where $S_{i_k}$ represents the volume of $\mathcal{S}(i_k)$.

---

**Algorithm 1** RWSADMM

---

1: **Initialization:**
   Initialize Markov transition matrices $\{\mathbf{P}(0), \mathbf{P}(1), \ldots, \}$.
   Initialize $\{\mathbf{x}_i^0\}_{i=1}^n = 0$, $\{\mathbf{z}_i^0\}_{i=1}^n = 0$, and

$$\mathbf{y}^1 = \frac{1}{n} \sum_{i=1}^n \left( \mathbf{x}_i^0 - \frac{\mathbf{z}_i^0}{\beta} \right) = 0 \quad (32)$$

2: **RWSADMM$(\beta, \mathbf{y}_1)$:**
3: **repeat**
4:     **for** $k \in 0, 1, 2, \ldots$ **do**
5:         Client $i_k$ receives $\mathbf{y}'_{i_k}$ and updates $\mathbf{X}$, $\mathbf{Z}$, and $\mathbf{y}$ using equations (11), (15), and (14);
6:     **end for**
       $\kappa = 0.99 \times \kappa$
7: **until** the termination condition is TRUE.
   **RETURN $\mathbf{X}^*, \mathbf{y}^*$**

---

## B    RWSADMM: Convergence Analysis

Our proof of convergence for the proposed stochastic ADMM-based federated learning algorithm is non-trivial and non-straightforward. It introduces novel techniques to address the challenge of integrating the mobilized server and stochasticity into our federated ADMM framework. Specifically, we carefully consider the movement of the server, which is realized by using a dynamic Markov matrix. This is a significant novelty and challenge in the proof, as it is the first method introduced in federated learning that considers this type of server movement. By introducing assumptions on the dynamic Markov matrix and accounting for the dynamic behavior of the server, we are able to guarantee the convergence of our algorithm under certain conditions. While there are a few existing works, such as [47] that address convergence rates and properties of LADMM algorithms, they are not directly applicable to our random walk mobilization and stochastic update setting and are not directly adaptable to the unique requirements of federated learning. As such, our work represents a significant contribution to the field and provides valuable insights into the convergence properties of stochastic ADMM-based federated learning algorithms. We believe that our novel approach and careful consideration of the movement of the server will inspire further research and development in this area, leading to more effective and efficient federated learning algorithms. The proof sketch of the Convergence Theorem (Theorem 4.8) is as follows.

*Proof.* The proof sketch is summarized as follows.

1. Under Assumption 4.2, the sequence created by the RWSADMM, i.e., $(\mathbf{y}^k, \mathbf{X}^k, \mathbf{Z}^k)_{k>T}$, satisfies
$$\left\| \mathbf{Z}^{k+1} - \mathbf{Z}^k \right\| = \sum \left\| \mathbf{z}_{i_k}^{k+1} - \mathbf{z}_{i_k}^k \right\| \leq L \left\| \mathbf{X}^{\tau(k,i_k)+1} - \mathbf{X}^{\tau(k,i_k)} \right\|$$

2. Recall $L_\beta^k$ defined in Eq. (25). For $k \geq 0$, the RWSADMM iterates satisfy
$$L_\beta^k - L_\beta \left( \mathbf{y}^{k+1}, \mathbf{X}^k; \mathbf{Z}^k \right) \geq \left\| \mathbf{y}^k - \mathbf{y}^{k+1} \right\|^2$$

3. Under Assumption 4.2, for $\beta > L$ and $\forall k > T$,

$$L_\beta(\mathbf{y}^{k+1}, \mathbf{X}^k; \mathbf{Z}^k) - L_\beta^{k+1} \geq \frac{\beta - L}{2n}\left\|\mathbf{X}^k - \mathbf{X}^{k+1}\right\|^2 - \frac{L^2}{n\beta}\left\|\mathbf{X}^{\tau(k,i_k)+1} - \mathbf{X}^{\tau(k,i_k)}\right\|^2$$

4. Recall $M_\beta^k$ defined in Eq. (25); under Assumption 4.2, for $\beta > 2L^2 + L + 2$ and $\forall k > T$,

$$M_\beta^k - M_\beta^{k+1} \geq \frac{\beta}{2}\left\|\mathbf{y}^k - \mathbf{y}^{k+1}\right\|^2 + \frac{1}{n}\left\|\mathbf{X}^k - \mathbf{X}^{k+1}\right\|^2 + \frac{L^2}{2n}\left\|\mathbf{X}^{\tau(k,i_k)+1} - \mathbf{X}^{\tau(k,i_k)}\right\|^2 \quad (33)$$

5. For $\beta > 2L^2 + L + 2$, RWSADMM ensures a lower bounded sequence $(M_\beta^k)_{k \geq 0}$.

$\square$

The proof details are provided in the following. Several steps are taken to prove Lemma 4.7. We introduce several Lemmas to represent these steps (Lemma B.1-B.5). Lemma B.1 shows that the update of the primal variable can bound the update on the dual variable.

**Lemma B.1.** Under Assumption 4.2, the sequence created by RWSADMM, $(\mathbf{y}^k, \mathbf{X}^k, \mathbf{Z}^k)_{k>T}$, satisfies,

$$\mathbb{E}\left\|\mathbf{Z}^{k+1} - \mathbf{Z}^k\right\| = \mathbb{E}\left\|\mathbf{z}_{i_k}^{k+1} - \mathbf{z}_{i_k}^k\right\| \leq L \cdot \mathbb{E}\left\|\mathbf{X}^{\tau(k,i_k)+1} - \mathbf{X}^{\tau(k,i_k)}\right\| \quad (34)$$

*Proof.* Note that client $i_k$ is activated at iteration $k$. Denote $\mathbf{x}_i^k$, $\mathbf{z}_i^k$, and $\mathbf{x}_i^k$ as the three groups of variables owned by any client $i$ ($1 \leq \forall i \leq n$) at iteration $k$. Under the Assumption 4.4, the optimality condition of $\mathbf{X}$ update for $i = i_k$ implies that

$$\mathbb{E}_\xi[sgn(\mathbf{t}')(g_i(\mathbf{x}_i^k, \xi) + \varepsilon_i^k - \mathbf{z}_i^k) + \beta(\mathbf{y}^{k+1} - \mathbf{x}_{i_k}^{k+1})] = 0 \quad (35)$$

Substituting Eq. (35) into Eq. (15) yields

$$\mathbb{E}_\xi[g_i(\mathbf{x}_i^k, \xi)] = \mathbb{E}_\xi[\mathbf{z}_{i_k}^{k+1}], \text{ for } i = i_k \quad (36)$$

Hence, for $i = i_k$, we have

$$\mathbb{E}\left\|\mathbf{z}_i^{k+1} - \mathbf{z}_i^k\right\| \overset{(a)}{=} \mathbb{E}\left\|\mathbf{z}_i^{k+1} - \mathbf{z}_i^{\tau(k,i)+1}\right\| \overset{(36)}{=} \left\|\mathbb{E}_\xi(g_i(\mathbf{x}_i^k, \xi) + \epsilon^{k+1}) - \mathbb{E}_\xi(g_i(\mathbf{x}_i^{\tau(k,i)}, \xi) + \epsilon^k)\right\|$$

$$\overset{(b)}{=} \left\|\mathbb{E}_\xi(g_i(\mathbf{x}_i^{\tau(k,i)+1}, \xi)) - \mathbb{E}_\xi(g_i(\mathbf{x}_i^{\tau(k,i)}, \xi))\right\| \overset{(17\&20)}{\leq} L \cdot \mathbb{E}_\xi\left\|\mathbf{x}_i^{\tau(k,i)+1} - \mathbf{x}_i^{\tau(k,i)}\right\| \quad (37)$$

where $\tau(k,i)$ is defined in Eq. (24). The equality $(a)$ holds because $\mathbf{z}_i^k = \mathbf{z}_i^{\tau(k,i)+1}$, and equality $(b)$ holds because $\mathbf{x}_i^k = \mathbf{x}_i^{\tau(k,i)+1}$. On the other hand, when $i \neq i_k$, agent $i$ is not activated at iteration $k$, so $\mathbb{E}_\xi[\|\mathbf{z}_{i_k}^{k+1} - \mathbf{z}_{i_k}^k\|] = L\mathbb{E}_\xi[\|\mathbf{x}_i^{k+1} - \mathbf{x}_i^k\|] = 0$. Therefore, we have the proof of Eq. (34). $\square$

Lemma B.2 shows that the $\mathbf{y}$-update in RWSADMM provides a sufficient descent of the augmented Lagrangian.

**Lemma B.2.** Recall $L_\beta^k$ defined in Eq. (25). For $k \geq 0$, RWSADMM iterates satisfies

$$L_\beta^k - L_\beta\left(\mathbf{y}^{k+1}, \mathbf{X}^k; \mathbf{Z}^k\right) \geq \left\|\mathbf{y}^k - \mathbf{y}^{k+1}\right\|^2 \quad (38)$$

*Proof.* We rewrite the augmented Lagrangian function $L$ in Eq. (8) by adding and subtracting the term $\|\mathbf{Z}\|^2/2\beta$ to the RHS of the equation and rearranging the terms. One can obtain

$$L_\beta(\mathbf{Y}, \mathbf{X}; \mathbf{Z}) = \frac{1}{n}\left(F(\mathbf{X}) + \frac{\beta}{2}\left\|\mathbf{1} \otimes \mathbf{y} - \mathbf{X}| - \boldsymbol{\epsilon} + \frac{\mathbf{Z}}{\beta}\right\|^2 - \frac{\|\mathbf{Z}\|^2}{2\beta}\right) \quad (39)$$

So, the augmented Lagrangian update is

$$L_\beta^k - L_\beta\left(\mathbf{y}^{k+1}, \mathbf{X}^k; \mathbf{Z}^k\right) = \frac{\beta}{2n}\left\|\left|\mathbf{1}\otimes\mathbf{y}^k - \mathbf{X}^k\right| - \boldsymbol{\epsilon}^k + \frac{\mathbf{Z}^k}{\beta}\right\|^2 - \frac{\beta}{2n}\left\|\left|\mathbf{1}\otimes\mathbf{y}^{k+1} - \mathbf{X}^k\right| - \boldsymbol{\epsilon}^k + \frac{\mathbf{Z}^k}{\beta}\right\|^2$$

$$\overset{(a)}{=} \frac{\beta}{2n}\sum_{i=1}^{n}\left(\left\|\mathbf{y}^k - \mathbf{y}^{k+1}\right\|^2 + 2\left\langle\left|\mathbf{y}^{k+1} - \mathbf{x}_i^k\right| - \boldsymbol{\epsilon}^k + \frac{\mathbf{z}_i^k}{\beta}, \mathbf{y}^k - \mathbf{y}^{k+1}\right\rangle\right) \geq \frac{\beta}{2}\left\|\mathbf{y}^k - \mathbf{y}^{k+1}\right\|^2 - \left\langle\mathbf{d}^k, \mathbf{y}^k - \mathbf{y}^{k+1}\right\rangle$$

$$(40)$$

The equality (a) is achieved by applying the cosine identity $\|b+c\|^2 - \|a+c\|^2 = \|b-a\|^2 + 2 < a+c, b-a >$, and $\mathbf{d}^k$ is defined as

$$\mathbf{d}^k := -\frac{\beta}{n}\sum_{i=1}^{n}\left(\left|\mathbf{y}^{k+1} - \mathbf{x}_i^k\right| - \boldsymbol{\epsilon}^k + \frac{\mathbf{z}_i^k}{\beta}\right) = 0 \tag{41}$$

which results from $\mathbf{y}-$update in the RWSADMM algorithm. $\square$

In Lemma B.3, the lower bound of descent in the augmented Lagrangian over the updates of $\mathbf{X}$ and $\mathbf{Z}$ is derived.

**Lemma B.3.** Recall $L_\beta^k$ defined in Eq. 25. Under Assumption 4.2, for $\beta > L$ and $\forall k > T$,

$$\mathbb{E}[L_\beta(\mathbf{y}^{k+1}, \mathbf{X}^k; \mathbf{Z}^k) - L_\beta^{k+1}] \geq \frac{\gamma - L}{2n}\mathbb{E}\left\|\mathbf{X}^k - \mathbf{X}^{k+1}\right\|^2 - \frac{L^2}{n\beta}\mathbb{E}\left\|\mathbf{X}^{\tau(k,i_k)+1} - \mathbf{X}^{\tau(k,i_k)}\right\|^2 \tag{42}$$

*Proof.* For the augmented Lagrangian $L_\beta$, we derive

$$L_\beta(\mathbf{y}^{k+1}, \mathbf{X}^k; \mathbf{Z}^k) - L_\beta^{k+1}$$

$$= \frac{1}{n}(f_{i_k}(\mathbf{x}_{i_k}^k) + \langle \mathbf{z}_{i_k}^k, |\mathbf{y}^{k+1} - \mathbf{x}_{i_k}^k| - \varepsilon_{i_k}^k \rangle + \frac{\beta}{2}\||\mathbf{y}^{k+1} - \mathbf{x}_{i_k}^k| - \varepsilon_{i_k}^k\|^2$$

$$- f_{i_k}(\mathbf{x}_{i_k}^{k+1}) - \langle \mathbf{z}_{i_k}^{k+1}, |\mathbf{y}^{k+1} - \mathbf{x}_{i_k}^{k+1}| - \varepsilon_{i_k}^k \rangle - \frac{\beta}{2}\||\mathbf{y}^{k+1} - \mathbf{x}_{i_k}^{k+1}| - \varepsilon_{i_k}^k\|^2)$$

$$= \frac{1}{n}(f_{i_k}(\mathbf{x}_{i_k}^k) - \langle \mathbf{z}_{i_k}^{k+1}, |\mathbf{y}^{k+1} - \mathbf{x}_{i_k}^{k+1}| - \varepsilon_{i_k}^k \rangle - f_{i_k}(\mathbf{x}_{i_k}^{k+1})$$

$$+ \langle \mathbf{z}_{i_k}^k, |\mathbf{y}^{k+1} - \mathbf{x}_{i_k}^k| - \varepsilon_{i_k}^k \rangle - \frac{\beta}{2}\left[\||\mathbf{y}^{k+1} - \mathbf{x}_{i_k}^{k+1}| - \varepsilon_{i_k}^k\|^2 - \||\mathbf{y}^{k+1} - \mathbf{x}_{i_k}^k| - \varepsilon_{i_k}^k\|^2\right])$$

$$\overset{(a)}{=} \frac{1}{n}(f_{i_k}(\mathbf{x}_{i_k}^k) - \langle \mathbf{z}_{i_k}^{k+1}, |\mathbf{y}^{k+1} - \mathbf{x}_{i_k}^{k+1}| - \varepsilon_{i_k}^k \rangle - f_{i_k}(\mathbf{x}_{i_k}^{k+1}) + \langle \mathbf{z}_{i_k}^k, |\mathbf{y}^{k+1} - \mathbf{x}_{i_k}^k| - \varepsilon_{i_k}^k \rangle$$

$$- \frac{\beta}{2}\left[\|\mathbf{x}_{i_k}^{k+1} - \mathbf{x}_{i_k}^k\|^2 + 2\langle |\mathbf{y}^{k+1} - \mathbf{x}_{i_k}^{k+1}| - \varepsilon_{i_k}, \mathbf{x}_{i_k}^{k+1} - \mathbf{x}_{i_k}^k \rangle\right])$$

$$= \frac{1}{n}(f_{i_k}(\mathbf{x}_{i_k}^k) - \langle \mathbf{z}_{i_k}^{k+1}, |\mathbf{y}^{k+1} - \mathbf{x}_{i_k}^{k+1}| - \varepsilon_{i_k}^k \rangle - f_{i_k}(\mathbf{x}_{i_k}^{k+1})$$

$$+ \langle \mathbf{z}_{i_k}^k, |\mathbf{y}^{k+1} - \mathbf{x}_{i_k}^k| - \varepsilon_{i_k}^k \rangle - \frac{\beta}{2}\|\mathbf{x}_{i_k}^{k+1} - \mathbf{x}_{i_k}^k\|^2$$

$$- \beta\langle |\mathbf{y}^{k+1} - \mathbf{x}_{i_k}^{k+1}| - \varepsilon_{i_k}, \mathbf{x}_{i_k}^{k+1} - \mathbf{x}_{i_k}^k \rangle)$$

$$\overset{(b)}{=} \frac{1}{n}(f_{i_k}(\mathbf{x}_{i_k}^k) - f_{i_k}(\mathbf{x}_{i_k}^{k+1}) + \langle \mathbf{z}_{i_k}^k, \frac{\mathbf{z}_{i_k}^{k+1} - \mathbf{z}_{i_k}^k}{\beta} + \mathbf{x}_{i_k}^{k+1} - \mathbf{x}_{i_k}^k \rangle - \frac{\beta}{2}\|\mathbf{x}_{i_k}^{k+1} - \mathbf{x}_{i_k}^k\|^2$$

$$- \langle \mathbf{z}_{i_k}^{k+1}, \frac{\mathbf{z}_{i_k}^{k+1} - \mathbf{z}_{i_k}^k}{\beta} \rangle - \beta\langle \frac{\mathbf{z}_{i_k}^{k+1} - \mathbf{z}_{i_k}^k}{\beta}, \mathbf{x}_{i_k}^{k+1} - \mathbf{x}_{i_k}^k \rangle)$$

$$= \frac{1}{n}(f_{i_k}(\mathbf{x}_{i_k}^k) + \frac{\mathbf{z}_{i_k}^{k+1}\mathbf{z}_{i_k}^k}{\beta} - \frac{(\mathbf{z}_{i_k}^k)^2}{\beta} + \mathbf{z}_{i_k}^k(\mathbf{x}_{i_k}^{k+1} - \mathbf{x}_{i_k}^k) - f_{i_k}(\mathbf{x}_{i_k}^{k+1})$$

$$+ \frac{\beta}{2}\|\mathbf{x}_{i_k}^k - \mathbf{x}_{i_k}^{k+1}\|^2 - \frac{(\mathbf{z}_{i_k}^{k+1})^2}{\beta} + \frac{\mathbf{z}_{i_k}^{k+1}\mathbf{z}_{i_k}^k}{\beta} - (\mathbf{z}_{i_k}^{k+1} - \mathbf{z}_{i_k}^k)(\mathbf{x}_{i_k}^k - \mathbf{x}_{i_k}^{k+1}))$$

$$= \frac{1}{n}(f_{i_k}(\mathbf{x}_{i_k}^k) - f_{i_k}(\mathbf{x}_{i_k}^{k+1}) - \frac{1}{\beta}(\mathbf{z}_{i_k}^k)^2 + (\mathbf{z}_{i_k}^{k+1})^2 - 2(\mathbf{z}_{i_k}^{k+1}\mathbf{z}_{i_k}^k)$$

$$+ \frac{\beta}{2}\|\mathbf{x}_{i_k}^k - \mathbf{x}_{i_k}^{k+1}\|^2 - \mathbf{z}_{i_k}^{k+1}(\mathbf{x}_{i_k}^k - \mathbf{x}_{i_k}^{k+1}))$$

$$\overset{(c)}{=} \frac{1}{n}(f_{i_k}(\mathbf{x}_{i_k}^k) + \frac{\beta}{2}\|\mathbf{x}_{i_k}^k - \mathbf{x}_{i_k}^{k+1}\|^2 - f_{i_k}(\mathbf{x}_{i_k}^{k+1}) - \frac{1}{\beta}\|\mathbf{z}_{i_k}^{k+1} - \mathbf{z}_{i_k}^k\|^2 - \langle \mathbf{z}_{i_k}^{k+1}, \mathbf{x}_{i_k}^k - \mathbf{x}_{i_k}^{k+1} \rangle)$$

$$\tag{43}$$

Where equality (a) holds due to the cosine identity $\|b + c\|^2 - \|a + c\|^2 = \|b - a\|^2 + 2 < a + c, b - a >$, equality (b) holds because of $\mathbf{y}-$update in RWSADMM, and equality (c) holds due to recursion of $\mathbf{y}-$update in RWSADMM. Next, we apply the stochastic property to Eq. (43) using Eq. (36),

$$\mathbb{E}[L_\beta(\mathbf{y}^{k+1}, \mathbf{X}^k; \mathbf{Z}^k) - L_\beta^{k+1}]$$

$$= \frac{1}{n}(\mathbb{E}_\xi[f_{i_k}(\mathbf{x}_{i_k}^k) - f_{i_k}(\mathbf{x}_{i_k}^{k+1})] - < E_\xi(g_i(\mathbf{x}_i^k, \xi) + \epsilon^k), \mathbf{x}_{i_k}^k - \mathbf{x}_{i_k}^{k+1} >$$

$$+ \frac{\beta}{2}E_\xi\|\mathbf{x}_{i_k}^k - \mathbf{x}_{i_k}^{k+1}\|^2 - \frac{1}{\beta}\mathbb{E}_\xi\|\mathbf{z}_{i_k}^{k+1} - \mathbf{z}_{i_k}^k\|^2)$$

$$\overset{(a)}{\geq} \frac{1}{n}\left(-\frac{L}{2}\mathbb{E}_\xi\|\mathbf{x}_{i_k}^k - \mathbf{x}_{i_k}^{k+1}\|^2 + \frac{\beta}{2}\mathbb{E}_\xi\|\mathbf{x}_{i_k}^k - \mathbf{x}_{i_k}^{k+1}\|^2 - \frac{1}{\beta}\mathbb{E}_\xi\|\mathbf{z}_{i_k}^{k+1} - \mathbf{z}_{i_k}^k\|^2\right) \qquad (44)$$

$$\overset{(b)}{\geq} \frac{1}{n}(-\frac{L}{2}\mathbb{E}_\xi\|\mathbf{x}_{i_k}^k - \mathbf{x}_{i_k}^{k+1}\|^2 + \frac{\beta}{2}\mathbb{E}_\xi\|\mathbf{x}_{i_k}^k - \mathbf{x}_{i_k}^{k+1}\|^2 - \frac{L^2}{\beta}\mathbb{E}_\xi\|\mathbf{x}_{i_k}^{\tau(k,i_k)+1} - \mathbf{x}_{i_k}^{\tau(k,i_k)}\|^2)$$

$$= \frac{\beta - L}{2n}\mathbb{E}_\xi\|\mathbf{x}_{i_k}^k - \mathbf{x}_{i_k}^{k+1}\|^2 - \frac{L^2}{n\beta}\mathbb{E}_\xi\|\mathbf{x}_{i_k}^{\tau(k,i_k)+1} - \mathbf{x}_{i_k}^{\tau(k,i_k)}\|^2$$

Where inequality (a) is achieved using Assumption 4.2 (Eq. (18)), and inequality (b) is due to Lemma B.1 (Eq. (34)). □

In Lemma B.4, the sufficient descent in Lyapunov functions is established.

**Lemma B.4.** Recall $M_\beta^k$ defined in Eq. (25); under Assumption 4.2, for $\beta > 2L^2 + L + 2$ and $\forall k > T$,

$$M_\beta^k - M_\beta^{k+1} \geq \frac{\beta}{2}\|\mathbf{y}^k - \mathbf{y}^{k+1}\|^2 + \frac{1}{n}\|\mathbf{X}^k - \mathbf{X}^{k+1}\|^2 + \frac{L^2}{2n}\|\mathbf{X}^{\tau(k,i_k)+1} - \mathbf{X}^{\tau(k,i_k)}\|^2 \qquad (45)$$

*Proof.* Using Eq. (25), we can attain

$$M_\beta^k - M_\beta^{k+1}$$

$$= L_\beta^k - L_\beta^{k+1} + \frac{L^2}{n}(\|\mathbf{x}_i^{\tau(k,i_k)+1} - \mathbf{x}_i^{\tau(k,i_k)}\|^2 - \|\mathbf{x}_i^{\tau(k+1,i_k)+1} - \mathbf{x}_i^{\tau(k+1,i_k)}\|^2)$$

$$\overset{(a)}{=} L_\beta^k - L_\beta^{k+1} + \frac{L^2}{n}(\|\mathbf{x}_{i_k}^{\tau(k,i_k)+1} - \mathbf{x}_{i_k}^{\tau(k,i_k)}\|^2 - \|\mathbf{x}_{i_k}^{k+1} - \mathbf{x}_{i_k}^k\|^2) \qquad (46)$$

$$= L_\beta^k - L_\beta^{k+1} + \frac{L^2}{n}(\|\mathbf{X}^{\tau(k,i_k)+1} - \mathbf{X}^{\tau(k,i_k)}\|^2 - \|\mathbf{X}^{k+1} - \mathbf{X}^k\|^2)$$

where inequality (a) is due to the following property,

$$x_i^{\tau(k+1,i)+1} - x_i^{\tau(k+1,i)} = \begin{cases} x_i^{k+1} - x_i^k, & i = i_k \\ x_i^{\tau(k,i)+1} - x_i^{\tau(k,i)}, & otherwise \end{cases} \qquad (47)$$

Substituting Eq. (33), one can obtain

$$M_\beta^k - M_\beta^{k+1} \geq L_\beta\left(\mathbf{y}^{k+1}, \mathbf{X}^k; \mathbf{Z}^k\right) - L_\beta^{k+1} + \|\mathbf{y}^k - \mathbf{y}^{k+1}\|^2 + \frac{L^2}{n}\left(\|\mathbf{X}^{\tau(k,i_k)+1} - \mathbf{X}^{\tau(k,i_k)}\|^2 - \|\mathbf{X}^{k+1} - \mathbf{X}^k\|^2\right) \qquad (48)$$

One can also substitute Eq. (33) into Eq. (48), which leads to

$$M_\beta^k - M_\beta^{k+1}$$

$$\geq \frac{\beta - L}{2n}\|\mathbf{X}^k - \mathbf{X}^{k+1}\|^2 - \frac{L^2}{n\beta}\|\mathbf{X}^{\tau(k,i_k)+1} - \mathbf{X}^{\tau(k,i_k)}\|^2$$

$$+ \|\mathbf{y}^k - \mathbf{y}^{k+1}\|^2 + \frac{L^2}{n}(\|\mathbf{X}^{\tau(k,i_k)+1} - \mathbf{X}^{\tau(k,i_k)}\|^2 - \|\mathbf{X}^{k+1} - \mathbf{X}^k\|^2) \qquad (49)$$

$$= \frac{\beta - L - 2L^2}{2n}\|\mathbf{X}^k - \mathbf{X}^{k+1}\|^2 + \frac{L^2(\beta - 1)}{n\beta}\|\mathbf{X}^{\tau(k,i_k)+1} - \mathbf{X}^{\tau(k,i_k)}\|^2 + \|\mathbf{y}^k - \mathbf{y}^{k+1}\|^2$$

Using $\frac{\beta}{2} - \frac{L}{2} - L^2 \geq 1$, $1 - \frac{1}{\beta} > \frac{1}{2}$, and $\beta < 2$, we complete the proof of Eq. (45). □

Lemma B.5 states that the Lyapunov function $M_\beta^k$ is lower bounded.

**Lemma B.5.** For $\beta > 2L^2 + L + 2$, RWSADMM ensures a lower bounded sequence $(M_\beta^k)_{k \geq 0}$ in expectation.

*Proof.* For $k > T$, we have

$$
\mathbb{E}[M_\beta^k]
$$

$$
= \mathbb{E}[L_\beta^k] + \frac{L^2}{n} \sum_{i=1}^n \mathbb{E}\left\| \mathbf{x}_i^{\tau(k,i_k)+1} - \mathbf{x}_i^{\tau(k,i_k)} \right\|^2
$$

$$
= \frac{1}{n} \sum_{j=1}^n \mathbb{E}\left( f_j(\mathbf{x}_j^k) + \left\langle \mathbf{z}_j^k, |\mathbf{y}^k - \mathbf{x}_j^k| - \varepsilon_j^k \right\rangle \right) + \frac{\beta}{2n} \sum_{j=1}^n \mathbb{E}\left\| \mathbf{y}_j^k - \mathbf{X}_j^k \right\|^2
$$

$$
+ \frac{L^2}{n} \sum_{i=1}^n \mathbb{E}\left\| \mathbf{x}_i^{\tau(k,i_k)+1} - \mathbf{x}_i^{\tau(k,i_k)} \right\|^2
$$

$$
\overset{(a)}{=} \frac{1}{n} \sum_{j=1}^n \mathbb{E}\left( f_j(\mathbf{x}_j^k) + \left\langle \mathbb{E}_\xi(g_j(\mathbf{x}_j^{\tau(k,j)}, \xi)), \mathbf{y}^k - \mathbf{x}_j^k \right\rangle \right) + \frac{\beta}{2n} \sum_{j=1}^n \mathbb{E}\left\| \mathbf{y}_j^k - \mathbf{X}_j^k \right\|^2 + \frac{L^2}{n} \sum_{i=1}^n \mathbb{E}\left\| \mathbf{x}_i^k - \mathbf{x}_i^{\tau(k,i_k)} \right\|^2
$$

$$
\overset{(b)}{\geq} \frac{1}{n} \sum_{j=1}^n \mathbb{E}\left( f_j(\mathbf{y^k}) + \left\langle \mathbb{E}_\xi(g_j(\mathbf{x}_j^{\tau(k,j)}, \xi)) - \mathbb{E}_\xi(g_j(\mathbf{x}_i^k, \xi)), \mathbf{y}^k - \mathbf{x}_j^k \right\rangle \right)
$$

$$
+ \frac{\beta - L}{2n} \sum_{j=1}^n \mathbb{E}\left\| \mathbf{y}_j^k - \mathbf{X}_j^k \right\|^2 + \frac{L^2}{n} \sum_{i=1}^n \mathbb{E}\left\| \mathbf{x}_i^k - \mathbf{x}_i^{\tau(k,i_k)} \right\|^2
$$

$$
\overset{(c)}{\geq} \frac{1}{n} \sum_{j=1}^n \mathbb{E}\left( f_j(\mathbf{y^k}) + \left\| \mathbb{E}_\xi(g_j(\mathbf{x}_i^{\tau(k,j)}, \xi)) - \mathbb{E}_\xi(g_j(\mathbf{x}_i^k, \xi) \right\|^2 \right)
$$

$$
+ \frac{\beta - L - 2}{2n} \mathbb{E}\left\| \mathbf{1} \otimes \mathbf{y}^k - \mathbf{X}^k \right\|^2 + \frac{L^2}{n} \sum_{i=1}^n \mathbb{E}\left\| \mathbf{x}_i^k - \mathbf{x}_i^{\tau(k,i_k)} \right\|^2
$$

$$
\overset{(d)}{\geq} \min_{\mathbf{y}} \left\{ \frac{1}{n} \sum_{j=1}^n \mathbb{E} f_j(\mathbf{y}^k) \right\} + \frac{2L^2}{2n} \mathbb{E}\left\| \mathbf{1} \otimes \mathbf{y}^k - \mathbf{X}^k \right\|^2
$$

$$
\overset{(e)}{\geq} -\infty
$$

(50)

where (a) holds due to Eq. (36). (b) holds because $f_j$ is Lipschitz differentiable (Eq. (22)). (c) holds due to Young's inequality. (d) follows from the assumption $\beta > 2L^2 + L + 2$ and the Lipschitz smoothness of each $f_j$, and (e) holds due to Assumption 4.1. Therefore, $M_\beta^k$ is bounded from below in expectation. $\square$

Now we can prove Lemma 4.7 with the above lemmas.

*Proof.* Recall that the maximum hitting time $T$ is almost surely finite. For Statement 1, the monotonicity of Lyapunov function $(M_\beta^k)_{k>T}$ in Lemma B.4 and their lower boundedness in Lemma B.5 ensure convergence of $(M_\beta^k)_{k \geq 0}$. For Statement 2, consider Statement 1 and the lower boundedness of $(M_\beta^k)_{k>T}$ in Lemma B.5 (Eq. (50)). $\frac{1}{n} F(\mathbf{y}^k)$ is upper bounded by $\max\{ \max_{t \in \{0,\ldots,T\}} \{\frac{1}{n} F(\mathbf{y}^t)\}, M_\beta^{T+1} \}$; and $\left\| \mathbf{1} \otimes \mathbf{y}_j^k - \mathbf{X}_j^k \right\|^2$ is upper bounded by $\max\{ \max_{\{t \in 0,\ldots,T\}} \{\|\mathbf{1} \otimes \mathbf{y}_j^t - \mathbf{X}_j^t\|^2\}, L_\beta^{T+1} \}$. By Assumption 4.1, the sequence $\{\mathbf{y}|k = \{0, 1, \ldots, \}\}$ is bounded. The boundedness of $\left\| \mathbf{1} \otimes \mathbf{y}_j^k - \mathbf{X}_j^k \right\|^2$ further leads to that of $\{\mathbf{X}_j^k|k = \{0, 1, \ldots, \}\}$. Finally, Eq. (36) and Assumption 4.2 ensure $(\mathbf{Z}^k)$ is bounded as well. Altogether, $(\mathbf{y}^k, \mathbf{X}^k, \mathbf{Z}^k)$ is bounded. $\square$

Based on Lemma 4.7, the convergence of the subgradients of $L_\beta^k$ can be established as follows.

**Lemma B.6.** With Assumption 4.5 and $\beta$ given in Lemma 4.7, for any given subsequence (including the whole sequence) with its index $(k_s)_{s\geq 0}$, there exists a sequence $(g^k)_{k\geq 0}$ with $(g^k) \in \partial L_\beta^{k+1}$ containing an almost surely convergent subsequence $(g^{k_{s_j}})_{j\geq 0}$, that is,

$$Pr\left(\lim_{j\to\infty}\left\|g^{k_{s_j}} = 0\right\|\right) = 1$$

*Proof.* The proof sketch is summarized as follows.

1. We construct the sequence $g^k \in \partial L_\beta^{k+1}$ and show that its subvector $q_i^k := (g_{\mathbf{y}^i}^k, g_{\mathbf{x}^i}^k, g_{\mathbf{z}^i}^k)$ satisfies $\lim_{k\to\infty} \mathbb{E}\left\|q_{i_k}^{k-\tau(\delta)-1}\right\|^2 = 0$, where the mixing time $\tau(\delta)$ is defined in Eq. (6).

2. For $k \geq 0$, define the filtration of sigma algebras:
   $\chi^k = \sigma\left(\mathbf{y}^0, \ldots, \mathbf{y}^k, \mathbf{X}^0, \ldots, \mathbf{X}^k, \mathbf{Z}^0, \ldots, \mathbf{Z}^k, i_0, \ldots, i_k\right)$. We show that

   $$\mathbb{E}\left(\left\|q_{i_k}^{k-\tau(\delta)-1}\right\|^2 \middle| \chi^{k-\tau(\delta)}\right) \geq (1-\delta)\pi_*\left\|g^{k-\tau(\delta)-1}\right\|^2,$$

   where $\pi_*$ is the minimal value in the Markov chain's stationary distribution. From this bound and the result in Step 1, we can get $\lim_{k\to\infty}\left\|g^k\right\| = 0$.

3. From the result in Step 2, we use the Borel-Cantelli lemma [48] to obtain an almost unquestionably convergent subsubsequence of $g^k$.

$\square$

The details of these steps are given as follows.

*Proof.* First, recall Lemma B.4 and $T < \infty$, we have

$$\sum_{k=0}^{\infty}\left(\mathbb{E}\left\|\mathbf{y}^k - \mathbf{y}^{k+1}\right\|^2 + \mathbb{E}\left\|\mathbf{X}^k - \mathbf{X}^{k+1}\right\|^2 + \mathbb{E}\left\|\mathbf{X}^{\tau(k,i_k)} - \mathbf{X}^{\tau(k,i_k)+1}\right\|_2^2\right) < +\infty \tag{51}$$

Hence, by Lemma B.4, one can infer

$$\sum_{k=0}^{\infty}\left(\mathbb{E}\left\|\mathbf{y}^k - \mathbf{y}^{k+1}\right\|^2 + \mathbb{E}\left\|\mathbf{X}^k - \mathbf{X}^{k+1}\right\|^2 + \mathbb{E}\left\|\mathbf{Z}^k - \mathbf{Z}^{k+1}\right\|_2^2\right) < +\infty \tag{52}$$

**Step 1:** The proof starts with computing the subgradients of the augmented Lagrangian Eq. (39) with the updates in RWSADMM,

$$\frac{\partial L_\beta^{k+1}}{\partial \mathbf{y}} \ni -\frac{\beta}{n}(\mathbf{x}_{i_k}^{k+1} - \mathbf{x}_{i_k}^k) + \frac{1}{n}(\mathbf{z}_{i_k}^{k+1} - \mathbf{z}_{i_k}^k) =: w^k \tag{53}$$

$$\nabla_{\mathbf{x}_j} L_\beta^{k+1} = \frac{1}{n}\left(\nabla f_j(\mathbf{x}_j^{k+1}) - \mathbf{z}_j^{k+1} + \beta(\mathbf{x}_j^{k+1} - \mathbf{y}^{k+1})\right) \tag{54}$$

$$\nabla_{\mathbf{z}_j} L_\beta^{k+1} = \frac{1}{n}(\mathbf{y}^{k+1} - \mathbf{x}_j^{k+1}) \tag{55}$$

We define $g^k$ and $q_i^k$ as

$$g^k := \begin{bmatrix} w^k \\ \nabla_{\mathbf{X}} L_\beta^{k+1} \\ \nabla_{\mathbf{Z}} L_\beta^{k+1} \end{bmatrix}, q_i^k := \begin{bmatrix} w^k \\ \nabla_{\mathbf{x}_i} L_\beta^{k+1} \\ \nabla_{\mathbf{z}_i} L_\beta^{k+1} \end{bmatrix} \tag{56}$$

where $i \in \mathbf{V}$ is the index of the agent and $g^k$ is the gradient of $L_k^\beta$. For $\delta \in (0, 1)$ and $k \geq \tau(\delta) + 1$,

$$\left\| q_{i_k}^{k-\tau(\delta)-1} \right\|^2 = \left\| q_{i_k}^{k-\tau(\delta)-1} - q_{i_k}^k + q_{i_k}^k \right\|^2 \overset{(a)}{\leq} 2\underbrace{\left\| q_{i_k}^{k-\tau(\delta)-1} - q_{i_k}^k \right\|^2}_{A} + 2\underbrace{\left\| q_{i_k}^k \right\|^2}_{B} \quad (57)$$

We upper bound A and B separately. A has three parts corresponding to the three parts of $g$. Its first part is

$$
\begin{aligned}
&\left\| w^{k-\tau(\delta)-1} - w^k \right\|^2 \\
&\leq 2\left\| w^{k-\tau(\delta)-1} \right\|^2 + 2\left\| w^k \right\|^2 \\
&\overset{(53)}{\leq} \frac{4}{n^2}\left( \beta^2 \left\| \mathbf{X}^{k+1} - \mathbf{X}^k \right\|^2 + \beta^2 \left\| \mathbf{X}^{k-\tau(\delta)} - \mathbf{X}^{k-\tau(\delta)-1} \right\|^2 \right. \\
&\left. + \left\| \mathbf{Z}^{k+1} - \mathbf{Z}^k \right\|^2 + \left\| \mathbf{Z}^{k-\tau(\delta)} - \mathbf{Z}^{k-\tau(\delta)-1} \right\|^2 \right)
\end{aligned}
\quad (58)
$$

Then by Eq. (54), we bound the 2nd part of $A$

$$
\begin{aligned}
&\left\| \boldsymbol{\nabla}_{\mathbf{x}_{i_k}} L_\beta^{k-\tau(\delta)-1} - \boldsymbol{\nabla}_{\mathbf{x}_{i_k}} L_\beta^{k+1} \right\|^2 \\
&\overset{(a)}{\leq} \frac{4L^2 + 4\beta^2}{n^2}\left\| \mathbf{x}_{i_k}^{k-\tau(\delta)-1} - \mathbf{x}_{i_k}^{k+1} \right\|^2 + \frac{4}{n^2}\left\| \mathbf{z}_{i_k}^{k-\tau(\delta)-1} - \mathbf{z}_{i_k}^{k+1} \right\|^2 \\
&+ \frac{4\beta^2}{n^2}\left\| \mathbf{y}^{k-\tau(\delta)-1} - \mathbf{y}^{k+1} \right\|^2 \\
&\leq \frac{D}{n^2}\sum_{t=k-\tau(\delta)-1}^{k}\left( \left\| \mathbf{y}^t - \mathbf{y}^{t+1} \right\|^2 + \left\| \mathbf{X}^t - \mathbf{X}^{t+1} \right\|^2 + \left\| \mathbf{Z}^t - \mathbf{Z}^{t+1} \right\|^2 \right)
\end{aligned}
\quad (59)
$$

where $D = (\tau(\delta) + 2)(4 + 4\beta^2 + 4L^2)$, and (a) uses the inequality of arithmetic and geometric means and Lipschitz differentiability of $f_j$ in Assumption 4.2. From Eq. (55). The third part of $A$ can be bounded as

$$
\begin{aligned}
&\left\| \boldsymbol{\nabla}_{\mathbf{z}_{i_k}} L_\beta^{k-\tau(\delta)-1} - \boldsymbol{\nabla}_{\mathbf{z}_{i_k}} L_\beta^{k+1} \right\|^2 \\
&\leq \frac{2}{n^2}\left( \left\| \mathbf{y}^{k-\tau(\delta)-1} - \mathbf{y}^{k+1} \right\|^2 + \left\| \mathbf{x}_{i_k}^{k-\tau(\delta)-1} - \mathbf{x}_{i_k}^{k+1} \right\|^2 \right) \\
&\leq \frac{2(\tau(\delta) + 2)}{n^2}\sum_{t=k-\tau(\delta)-1}^{k}\left( \left\| \mathbf{y}^t - \mathbf{y}^{t+1} \right\|^2 + \left\| \mathbf{X}^t - \mathbf{X}^{t+1} \right\|^2 \right)
\end{aligned}
\quad (60)
$$

Plugging Eq. (58), Eq. (59), and Eq. (60) into term $A$, we get a constant $C_1 \sim O(\frac{\tau(\delta)+1}{n^2})$, depending on $\tau(\delta)$, $\beta$, $L$, and $n$, such that

$$A \leq C_1 \sum_{t=k-\tau(\delta)-1}^{k}\left( \left\| \mathbf{y}^t - \mathbf{y}^{t+1} \right\|^2 + \left\| \mathbf{x}^t - \mathbf{x}^{t+1} \right\|^2 + \left\| \mathbf{z}^t - \mathbf{z}^{t+1} \right\|^2 \right) \quad (61)$$

To bound the term $B$, using Eq. (55) and $\mathbf{Z}$-update (Eq. (15)), we have

$$\boldsymbol{\nabla}_{\mathbf{z}_{i_k}} L_\beta^{k+1} = \frac{1}{n\beta}(\mathbf{z}_{i_k}^{k+1} - \mathbf{z}_{i_k}^k) \quad (62)$$

Applying Eq. (54) and Eq. (36), we drive $\boldsymbol{\nabla}_{\mathbf{x}_{i_k}} L_\beta^{k+1}$:

$$\boldsymbol{\nabla}_{\mathbf{x}_{i_k}} L_\beta^{k+1} = \frac{1}{n\beta}\left( \boldsymbol{\nabla} f_{i_k}(\mathbf{x}_{i_k}^{k+1}) - \boldsymbol{\nabla} f_{i_k}(\mathbf{x}_{i_k}^k) + \mathbf{z}_{i_k}^k - \mathbf{z}_{i_k}^{k+1} \right) \quad (63)$$

So we have

$$B \leq C_2 \left( \left\| \mathbf{x}^t - \mathbf{x}^{t+1} \right\|^2 + \left\| \mathbf{z}^t - \mathbf{z}^{t+1} \right\|^2 \right) \tag{64}$$

for a constant $C_2$ depending on $L$, $\beta$, and $n$, in the order of $O(\frac{1}{n^2})$. Then substituting Eq. (61) and Eq. (64) into Eq. (57) and taking expectations, it yields

$$
\begin{aligned}
\mathbb{E} &\left\| q_{i_k}^{k-\tau(\delta)-1} \right\|^2 \\
&\leq C \sum_{t=k-\tau(\delta)-1}^{k} \left( \mathbb{E} \left\| \mathbf{y}^t - \mathbf{y}^{t+1} \right\|^2 + \mathbb{E} \left\| \mathbf{x}^t - \mathbf{x}^{t+1} \right\|^2 + \mathbb{E} \left\| \mathbf{z}^t - \mathbf{z}^{t+1} \right\|^2 \right)
\end{aligned}
\tag{65}
$$

where $C = C_1 + C_2$, and $C \sim O(\frac{\tau(\delta)+1}{n^2})$. Recalling Eq. (52), we get the convergence

$$\lim_{k \to \infty} \mathbb{E} \left\| q_{i_k}^{k-\tau(\delta)-1} \right\|^2 = 0 \tag{66}$$

which completes the proof of **Step 1**.

**Step 2:** We compute the expectation:

$$
\begin{aligned}
&\mathbb{E} \left( \left\| q_{i_k}^{k-\tau(\delta)-1} \right\|^2 \Big| \chi^{k-\tau(\delta)} \right) \\
&= \sum_{j=1}^{n} [\mathbf{P}(k)^{\tau(\delta)}]_{i_{k-\tau(\delta)},j} \left( \left\| \nabla_{\mathbf{y}} L_{\beta}^{k-\tau(\delta)} \right\|^2 + \left\| \nabla_{\mathbf{x_j}} L_{\beta}^{k-\tau(\delta)} \right\|^2 \right. \\
&\left. + \left\| \nabla_{\mathbf{z_j}} L_{\beta}^{k-\tau(\delta)} \right\|^2 \right) \overset{(a)}{\geq} (1-\delta) \pi_* \left\| g^{k-\tau(\delta)-1} \right\|^2
\end{aligned}
\tag{67}
$$

where (a) follows from Eq. (4) and the definition of $g_k$ in Eq. (56). Then, with Eq. (65), one can derive

$$\lim_{k \to \infty} \mathbb{E} \left\| g^k \right\|^2 = \lim_{k \to \infty} \mathbb{E} \left\| g^{k-\tau(\delta)-1} \right\|^2 = 0, \tag{68}$$

By the Schwarz inequality $(\mathbb{E} \| g^k \|)^2 \leq \mathbb{E} \| g^k \|^2$, we have

$$\lim_{k \to \infty} \mathbb{E} \| g^k \| = 0, \tag{69}$$

**Step 3:** By Markov's inequality, for each $\omega > 0$, it holds that

$$Pr(\|g_k\| > \omega) \leq \frac{\mathbb{E} \| g^k \|}{\omega} \overset{Eq.(68)}{\Longrightarrow} \lim_{k \to \infty} Pr(\|g_k\| > \omega) = 0 \tag{70}$$

when a subsequence $(k_s)_{s \geq 0}$ is provided, Eq. (69) implies,

$$Pr(\|g_{k_s}\| > \omega) = 0 \tag{71}$$

Then, for $j \in \mathbb{N}$, select $\omega = 2^{-j}$ and we can find a nondecreasing subsubsequence $(k_{s_j})$, such that

$$Pr(\|g_{k_s}\| > 2^{-j}) \leq 2^{-j}, \quad \forall k_s \geq k_{s_j} \tag{72}$$

We have

$$\sum_{j=1}^{\infty} Pr(\|g_{k_s}\| > 2^{-j}) \leq \sum_{j=1}^{\infty} 2^{-j} = 1 \tag{73}$$

The Borel-Cantelli lemma yields

$$Pr\left(\limsup_{j}\{\|g_{k_s}\| > 2^{-j}\}\right) = 0 \tag{74}$$

and thus

$$Pr\left(\lim_{j}\left\|g_{k_{s_j}}\right\| = 0\right) = 1 \tag{75}$$

This completes **Step 3** and thus the entire **Lemma B.6**.

$\square$

**Proof of Convergence**    Finally, we present the proof of the convergence theorem (Theorem 4.8) as follows.

*Proof.* By statement 2 of Lemma 4.7, the sequence $(\mathbf{y}^k, \mathbf{X}^k, \mathbf{Z}^k)$ is bounded, so there exists a convergent subsequence $(\mathbf{y}^{k_s}, \mathbf{X}^{k_s}, \mathbf{Z}^{k_s})$ converging to a limit point $(\mathbf{y}^*, \mathbf{X}^*, \mathbf{Z}^*)$ as $s \to \infty$. By continuity, we have

$$L_\beta(\mathbf{y}^*, \mathbf{X}^*, \mathbf{Z}^*) = \lim_{s \to \infty} L_\beta(\mathbf{y}^{k_s}, \mathbf{X}^{k_s}, \mathbf{Z}^{k_s}) \tag{76}$$

Lemma B.6 finds a subsubsequence $g^{k_{s_j}} \in \partial L_\beta^{k+1}$ such that $Pr\left(\lim_{j \to \infty}\left\|g^{k_{s_j}}\right\| = 0\right) = 1$. By the definition of general subgradient ([49], def 8.3), we have $0 \in \partial L_\beta(\mathbf{y}^*, \mathbf{X}^*, \mathbf{Z}^*)$. Hence, Theorem 4.8 is proved. $\square$

## C    RWSADMM: Convergence Rate Analysis

The detailed proof of the convergence rate theorem is as follows.

*Proof.* It can be verified that under specific initialization, Eq. (36) holds for all $k \geq 0$. Consequently, Lemmas B.1-B.4 hold for all $k \geq 0$. For $g^k$ defined in Eq. (56), Eq. (65) and Eq. (67) hold. Jointly applying Eq. (65) and Eq. (67), for any $k > \tau(\delta) + 1$, one has

$$\mathbb{E}\left\|g^{k-\tau(\delta)-1}\right\|^2 \leq \frac{C}{(1-\delta)\pi_*}$$
$$\sum_{t=k-\tau(\delta)-1}^{k}\left(\mathbb{E}\left\|\mathbf{y}^t - \mathbf{y}^{t+1}\right\|^2 + \mathbb{E}\left\|\mathbf{X}^t - \mathbf{X}^{t+1}\right\|^2 + \mathbb{E}\left\|\mathbf{Z}^t - \mathbf{Z}^{t+1}\right\|^2\right) \tag{77}$$

According to Lemmas B.1 and B.4, for $k \geq \tau(\delta) + 1$, it holds,

$$\sum_{t=k-\tau(\delta)-1}^{k}\left(\mathbb{E}\left\|\mathbf{y}^t - \mathbf{y}^{t+1}\right\|^2 + \mathbb{E}\left\|\mathbf{X}^t - \mathbf{X}^{t+1}\right\|^2 + \mathbb{E}\left\|\mathbf{Z}^t - \mathbf{Z}^{t+1}\right\|^2\right)$$
$$\leq \max\left\{\frac{2}{\beta - \gamma}, (1 + L^2)n\right\}\left(\mathbb{E}L_\beta^{k-\tau(\delta)-1} - \mathbb{E}L_\beta^{k+1}\right) \tag{78}$$

It implies that for any $k \geq 0$, it holds

$$\mathbb{E}\left\|g^k\right\|^2 \leq C'\left(\mathbb{E}L_\beta^k - \mathbb{E}L_\beta^{k+\tau(\delta)+2}\right) \tag{79}$$

where $C' := \max\{\frac{2}{\beta-\gamma}, (1 + L^2)n\}\frac{C}{(1-\delta)\pi_*}$. It can be verified that $C' = O(\frac{\tau(\delta)+1}{(1-\delta)n\pi_*})$. Let $\tau' := \tau(\delta) + 2$; for any $K > \tau'$, summing Eq. (79) over $k \in \{K - \tau', \ldots, \mathrm{mod}_{\tau'}K\}$ gives

$$\sum_{l=1}^{\lfloor \frac{K}{\tau'} \rfloor} \mathbb{E}\left\|g^{K-l\tau'}\right\|^2 \leq C'\left(\mathbb{E}L_{\beta}^{\mathrm{mod}_{\tau'}K} - \mathbb{E}L_{\beta}^{K}\right) \leq C'(L_{\beta}^0 - \underline{f}) \tag{80}$$

where the last inequality follows from the non-decreasing property of the sequence $(L_{\beta}^k)_{k\geq 0}$ and the fact that $(L_{\beta}^k)_{k\geq 0}$ is lower bounded by $\underline{f}$. According to Eq. (80),

$$\begin{aligned}
\min_{k\leq K}\mathbb{E}\left\|g^k\right\|^2 &\leq \min_{1\leq l\leq \lfloor \frac{K}{\tau'} \rfloor} \mathbb{E}\left\|g^{K-l\tau'}\right\|^2 \\
&\leq \frac{1}{\lfloor \frac{K}{\tau'} \rfloor}\sum_{l=1}^{\lfloor \frac{K}{\tau'} \rfloor}\mathbb{E}\left\|g^{K-l\tau'}\right\|^2 \leq \frac{\tau'C'}{K-\tau'}(L_{\beta}^0 - \underline{f}) \\
&\leq \frac{C'(\tau'+1)}{K}(L_{\beta}^0 - \underline{f})
\end{aligned} \tag{81}$$

where the constant $C'(\tau'+1) = O\left(\frac{\tau(\delta)^2+1}{(1-\delta)n\pi_*}\right)$. $\hfill\square$

We consider a reversible Markov chain on an undirected graph. Using the definition of $\tau(\delta)$ in Eq. (6) and setting $\delta = 1/2$, one has $\tau(\delta) \sim \frac{\ln n}{1-\lambda_2(P(k))}$. To guarantee $\min_{k\leq K}\mathbb{E}\left\|g^k\right\|^2 \leq \omega$, certain number of iterations is sufficient,

$$O\left(\frac{1}{\omega}\cdot\left(\frac{\ln^2 n}{(1-\lambda_2(P(k)))^2}\right) + 1\right) \tag{82}$$

## D  Experiments

### D.1  Models

One strongly convex model and two non-convex models are utilized in our implementations. An MLR model with a logistic regression classifier is implemented for the strongly convex setting. For the first non-convex setting, an MLP model with two hidden dense layers of vector image size (resizing the 2D image as 1D vector) and 100 hidden nodes in the hidden layer are implemented. The cross-entropy loss is employed for this network. For the second non-convex setting, a CNN model with two convolutional layers with convolution operations of size $5 \times 5$ and one fully-connected layer of 512 followed by a Softmax layer is implemented. The cross-entropy loss is utilized in the network, and dropout rates of 25% and 50% are applied after convolutional layers.

### D.2  Graph Construction

To meet this requirement on Assumption 3.1, we propose using a Markov transition matrix $\mathbf{P}$ with a maximum eigenvalue of less than $1 - 1/m^{2/3}$, where $m$ is the number of edges. To fulfill this inequality, we can increase the number of edges in the network. In our experiments, we have addressed this issue by requiring each client to be a neighbor of at least $M$ other clients, which ensures a sufficient number of edges to satisfy the assumption on $\mathbf{P}$. By incorporating this requirement into our implementation, we can guarantee the validity of our results and ensure that our algorithm performs optimally under realistic conditions.

### D.3  Hyperparameter tuning

The two hyperparameters of RWSADMM ($\beta$ and $\kappa$) must be fine-tuned to optimize the performance. In the first stage of the experiments, we set $\kappa = 0.001$ and search for the optimal values of $\beta$. The fine-tuning process is performed for each dataset and each model separately. In Fig. 3, the effect of $\beta$ values on the performance of RWSADMM for the MNIST dataset is presented.

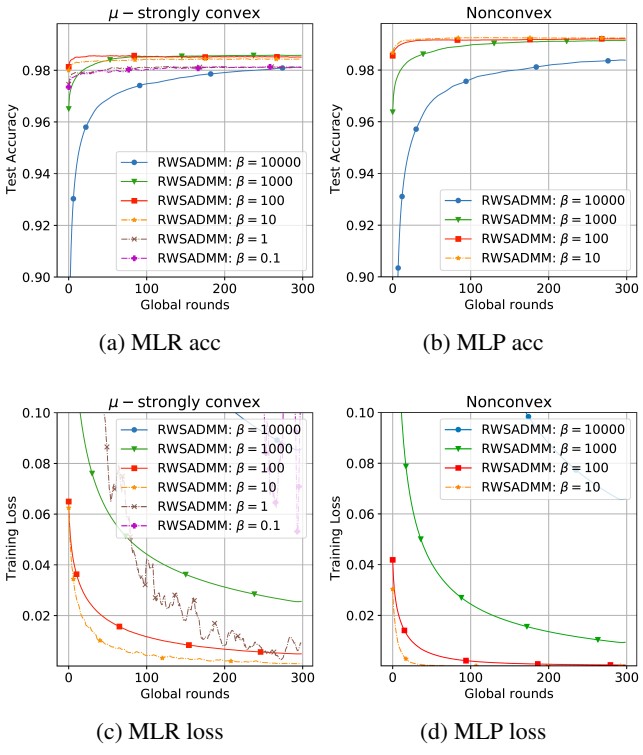

(a) MLR acc

(b) MLP acc

(c) MLR loss

(d) MLP loss

Figure 3: Effect of $\beta$ on the convergence of RWSADMM in the MLR (3a, 3c) and MLP (3b, 3d) models for MNIST dataset.

The $\kappa$ parameter, which affects the dual variable $\mathbf{Z}$, is also fine-tuned in the second stage. Same as the first stage, $\kappa$ is fine-tuned for the MNIST dataset for the MLR, MLP, and CNN models, and the results are shown in Fig. 4. The optimal values of $\{\kappa = 0.001, 0.01\}$ have the best performances for MLR and MLP, respectively.

The proximal parameter, $\epsilon$, is set to a fixed value of $\{\epsilon = 1e-5\}$ for all the experiments. For the MNIST dataset, the fine-tuned parameter values of $\beta = 10$ and $\kappa = 0.001$ for MLR, $\beta = 10$ and $\kappa = 0.01$ for MLP are used. For the Synthetic dataset, $\beta = 10$ and $\kappa = 0.01$ for MLR, $\beta = 100$ and $\kappa = 0.01$ for MLP models are utilized. Finally, for CIFAR10 dataset, $\beta = 100$ and $\kappa = 0.001$ for MLR, $\beta = 100$ and $\kappa = 1$ for MLP are used.

### D.4 CIFAR10 figures

The performance comparison of RWSADMM, FedAvg, Per-FedAvg, pFedMe, APFL, and Ditto for the Cifar10 dataset are depicted in Fig. 5. The fine-tuned values of $\beta = 0.001$ and $\kappa = 0.001$ are used for RWSADMM. The RWSADMM has a steep curve nearly reaching the optimal values from the first few rounds in strongly convex and non-convex models, indicating a faster convergence process than the benchmark algorithms. Also, RWSADMM shows a clear advantage for MLR or DNN models for accuracy.

### D.5 Synthetic figures

Due to the 1D nature of the synthetic dataset, only MLR and MLP models are utilized for it. The performance comparison of RWSADMM, FedAvg, Per-FedAvg, pFedMe, APFL, and Ditto for the Synthetic dataset are depicted in Fig. 6. The fine-tuned values of $\beta = 100$ and $\kappa = 0.001$ are used for RWSADMM for all the settings. By comparing the accuracy and loss diagrams, RWSADMM performs visibly better than the rest of the algorithms in both strongly convex and non-convex settings. The accuracy rate of RWSADMM shows accuracy improvement compared with the benchmark algorithms by the margin of $14\%$ for both MLR and MLP models.

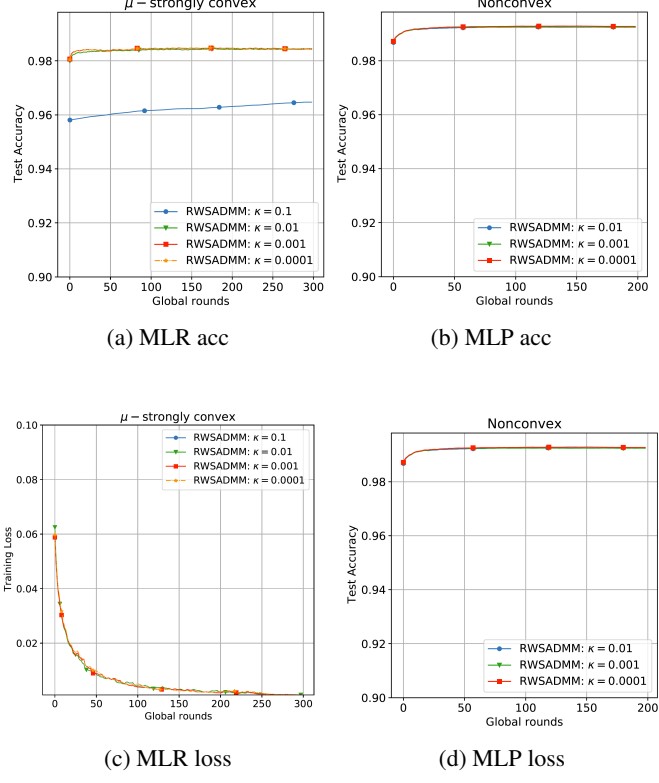

(a) MLR acc

(b) MLP acc

(c) MLR loss

(d) MLP loss

Figure 4: Effect of $\kappa$ on the convergence of RWSADMM in the MLR (4a, 4c) and MLP (4b, 4d) models for MNIST dataset.

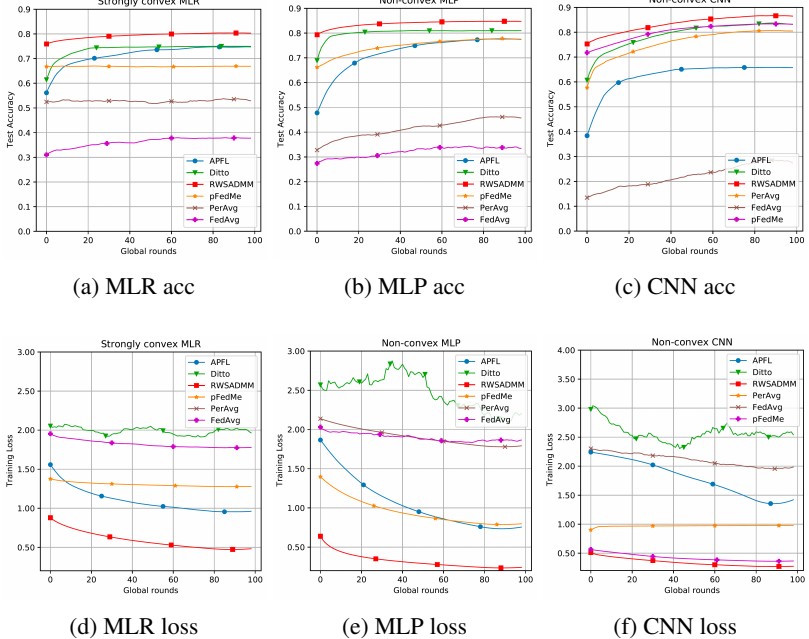

(a) MLR acc

(b) MLP acc

(c) CNN acc

(d) MLR loss

(e) MLP loss

(f) CNN loss

Figure 5: Performance comparison (test accuracy and training loss) of RWSADMM, pFedMe, Per-Avg, FedAvg, APFL, and Ditto for CIFAR10 dataset for strongly convex MLR, non-convex MLP, and non-convex CNN models.

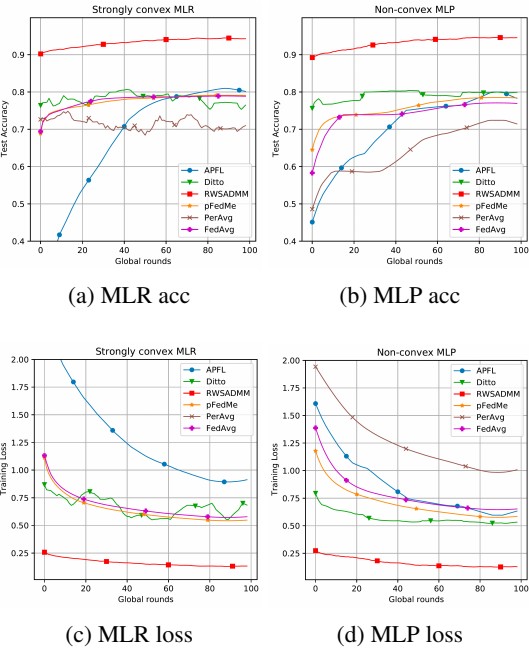

(a) MLR acc          (b) MLP acc

(c) MLR loss          (d) MLP loss

Figure 6: Performance comparison (test accuracy and training loss) of RWSADMM, pFedMe, Per-Avg, FedAvg, APFL, and Ditto for Synthetic dataset and different settings: strongly convex MLR and non-convex MLP.

## D.6 Different Number of Users

This appendix examines the effect of modifying the number of users/clients in the graph. The configuration values are all the same as the optional configuration for the RWSADMM algorithm with 20 agents. We keep the size of the neighborhood and the overall graph configurations the same for all the experiments. The batch size is decreased to 5 due to memory limitations, and the number of iterations is increased to 500. The total of users tested is 20, 50, and 100 users. The test accuracy and train loss progress curves for different numbers of clients are shown in Fig. 7.

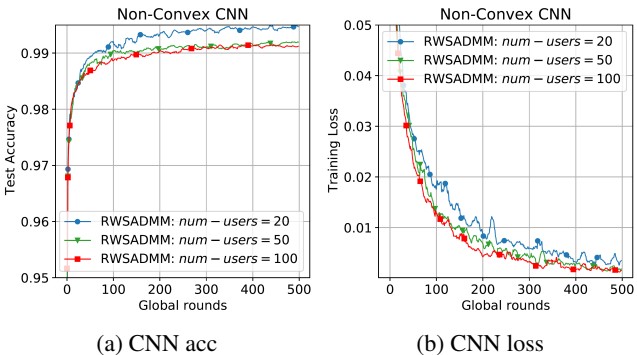

(a) CNN acc          (b) CNN loss

Figure 7: Performance comparison (test accuracy and training loss) of RWSADMM for different graphs with 20, 50, and 100 users/nodes in the graph.

As the number of users increases and the graph expands, the convergence gets more challenging, the test accuracy rates slightly decrease, and the time duration of the algorithm increases. The final test accuracy rates and time consumption of different graphs are presented in Table 2.

| RWSADMM | MNIST | |
| | CNN | |
| # of users | acc(%) | t(s) |
|---|---|---|
| 20 | 99.57 | 2929 |
| 50 | 99.25 | 6994 |
| 100 | 99.19 | 13878 |

Table 2: Test accuracy rate and time duration comparison of RWSADMM for different graphs with 20, 50, and 100 users/nodes in the graph.