# OpenReview forum: "Mobilizing Personalized Federated Learning in Infrastructure-Less and Heterogeneous Environments via Random Walk Stochastic ADMM"
_NeurIPS.cc/2023/Conference — NeurIPS 2023 poster_

### Official Review · Reviewer_cJVn · 2023-07-06

**Soundness:** 3 good
**Presentation:** 2 fair
**Contribution:** 2 fair
**Rating:** 4
**Confidence:** 4

**Summary:**

Authors consider in this paper a Federated Learning framework with a mobile server. They propose an algorithm named Random Walk Stochastic Alternating Direction Method of Multipliers (RWSADMM) for networks with instability and limited capacity. To overcome the non-IID nature of clients' data, they ensure local proximity among adjacent clients using hard inequality constraints. Their theoretical and empirical results demonstrate the effectiveness of RWSADMM.

**Strengths:**

The mobile Federated Learning in infrastructure-less environment is an important problem. The paper is well organized with rigorous theoretic analysis.

**Weaknesses:**

The random walk mobility model is over simplified and is not a realistic model for wireless adhoc network. The random walk model was initially developed to emulate the unpredictable movement of particles in physics. It is a memoryless mobility model that does not consider temporal and spatial correlation. In addition, topology control technique is a  must to ensure reliable operation of a wireless adhoc network in reality and therefore such a memoryless assumption is hardly the case in many real life applications.
Although the concept of the Federated Learning (FL) framework with a mobile server is interesting, it seems to be a special case of asynchronous FL, where only a subset of clients is selected to participate in the update in each iteration. Authors are recommended to elaborate the difference between this work and existing asynchronous FL work in the literature.

**Questions:**

Can authors elaborate more regarding employing the dynamic Markov Chain to model the server's movement?
The introduction of the y-variable in Eq. (7) is unclear.
If one of the neighboring nodes of the selected node in an iteration fails to respond, does the algorithm still work?

**Limitations:**

The mobility model considered in this paper is of limited applicability.

---

> ### Author Rebuttal · Authors · 2023-08-09
>
> We appreciate your diligent inquiry and thoughtful comments. In the subsequent sections, we will provide a detailed response to each of your questions.
>
> Weakness 1: model is over simplified:
>
> Answer 1: In our model, we enhance the conventional random walk by introducing a time-inhomogeneous Markov Probability Transition matrix denoted as P(k), where 'k' signifies the time index. Unlike the conventional fixed Probability Transition matrix 'P' in a standard random walk, our approach accommodates the dynamic movement patterns of nodes over time, rendering it better suited for real-world infrastructureless environments. Our proposed model functions through two iterative phases, both of which contribute to a more authentic portrayal of mobility patterns within the environment:
>
> Connectivity Graph Update: We revise the connectivity graph based on the node and battle vehicle positions. This step ensures that the potential destinations for the battle vehicle are confined to its connected nodes—those that are practically attainable or aligned with its specific route plan. This approach takes into account the limitations and constraints imposed by the mobility of the vehicle.
>
> Random Walk Algorithm with Accessible Nodes: The model employs the generalized random walk algorithm to choose a node from the set of accessible nodes guided by the transition matrix P(k). This selection process takes into consideration the spatial correlation between the battle vehicle and the nodes, resulting in more realistic mobility patterns that closely mirror the actual movement of the network components.
>
>
> Weakness 2: Elaborate the difference between this work and existing asynchronous FL work
>
> Answer 2:
> Infrastructureless Environment: Our RWSADMM framework is tailored to operate in scenarios where the network infrastructure is significantly inadequate, unreliable, or nonexistent. In contrast, many existing asynchronous FL methods, as pointed out in [r1][r2], assume a certain level of network infrastructure that guarantees timely model updates.This calls for certain level of network infrastructure that guarantees the timely model update. Whereas our RWSADMM framework excels precisely in situations where the network's capabilities are highly constrained, ensuring that convergence is achieved even without the need for deterministic or stochastic bounded delay constraints.
>
> Convergence in Challenging Conditions: Our method is designed to achieve convergence under conditions that are far more challenging than those typically encountered in traditional asynchronous FL. The incorporation of mobility patterns, dynamic changes in the environment, and the adaptability of our approach further distinguish our work from other asynchronous optimization strategies.
>
> [r1] Chen, Tianyi, et al. "Vafl: a method of vertical asynchronous federated learning." arXiv preprint arXiv:2007.06081 (2020).
>
> [r2] Xu, Chenhao, et al. "Asynchronous federated learning on heterogeneous devices: A survey." arXiv preprint arXiv:2109.04269 (2021).
>
>
> Q1: Elaborate more dynamic Markov Chain to model the server's movement?
>
> A1: In a dynamic and uncertain environment like a warzone, the server's movement cannot be assumed to always follow a fixed pattern or remain consistent over time. We employ a dynamic time-inhomogeneous Markov Chain to capture the time-varying nature of the server's movement. In this approach, the transition probabilities are allowed to change as a function of time or the current state, enabling us to model the server's mobility more realistically. The use of a non-homogeneous Markov Chain allows the server to adapt its movement pattern based on situational changes, terrain conditions, and tactical considerations.
> E.g., let us denote the state space of the Markov Chain as S = {s_1, s_2, ..., s_n}, where each state s_i represents a specific device in the warzone. The transition probability matrix at time t is denoted by P(t) = [p_ij(t)], where p_ij(t) represents the probability of transitioning from state s_i to state s_j at time t.
> Consider a simplified warzone scenario with three specific soldiers (clients): A, B, and C. Let the state space be S = {A, B, C}, representing the three clients. We initialize the transition probability matrix P(0) as follows:
> P(0) = | p_AA(0) p_AB(0) p_AC(0) |
> | p_BA(0) p_BB(0) p_BC(0) |
> | p_CA(0) p_CB(0) p_CC(0) |
> For instance, assume that the server starting from B has received new intelligence information indicating that client A currently moves to an unsafe or unreachable area due. This new information can be factored into updating the transition probabilities for the server's next movement.
> Let's assume we have the following updated transition probability matrix P(1) after considering the tactical considerations:
> P(1) = | p_AA(1) p_AB(1) p_AC(1) |
> | p_BA(1) p_BB(1) p_BC(1) |
> | p_CA(1) p_CB(1) p_CC(1) |
> In this case, we have modified the probabilities p_CA(1) and p_BA(1) to reflect a reduced likelihood of transitioning to location A from its current state. This reduction is a consequence of the tactical decision to avoid the route to A due to environmental concerns.
>
> Q2: The introduction of the y-variable in Eq. (7) is unclear.
>
> A2: The description of y is explained in lines 184 and 185, as y_i is stored on the server which is the local proximity of N(i) for client i. In other words, y_i is the local proximity of the local variables of agent i and its neighbors N(i).
>
> Q3: If one of the neighboring nodes of the selected node in an iteration fails to respond, does the algorithm still work?
>
> A3:If one of the neighboring nodes fails, the server can move on to proceed to aggregate model updates from the neighboring nodes that do respond successfully. Our mobilized FL framework allows for typical weighted aggregation of updates, where nodes that have communication failure can be assigned weights of 0 in the temporal aggregation process.

---

### Official Review · Reviewer_J9bJ · 2023-07-06

**Soundness:** 3 good
**Presentation:** 3 good
**Contribution:** 3 good
**Rating:** 7
**Confidence:** 3

**Summary:**

This paper proposes random walk stochastic method based on traditional ADMM, which enables personalized federated learning considering the mobility of clients and using a mobilized server. The authors solve the optimization of the stored client’s model parameters through Lagrangian and dual problem. The theoretical analysis proves the convergence performance of the proposed algorithm based on assumptions 4.1 to 4.5. The paper validates the performance of the algorithm by comparing its accuracy and loss with APFL/Ditto/PerAvg/FedAvg/pFedMe.

**Strengths:**

1.This paper realizes the mobilizing federated learning in a in an infrastructure-less environment for the first time.
2.This paper has rigorously proposed the RWSADMM algorithm and rigorously proves its convergence performance.
3.The paper solves the model update faster by solving the dual problem and achieves better communication cost.

**Weaknesses:**

1.The paper assume the mobilized server wanders in the region following a random walk pattern, and uses the warzone as an example. In a drastic combat, the mobilized server should usually follow a more regularized route instead of randomly wandering in the field. Also, the client are also set to be stationary in the paper, which can not represent usual moving soldiers and devices. Therefore, I think the system model is not very realistic, and the applicability of the system in a real war may be limited.

**Questions:**

1.On page 8, the authors claim RWSADMM’s O(\omega^{-1}) communication complexity is better than Per-FedAvg’s O(\omega^{-3/2}) complexity. This hold when \omega ranges from 0 to 1, but the authors seem not to discuss or define the range of the parameter.

---

> ### Author Rebuttal · Authors · 2023-08-09
>
> We sincerely value your positive review of our work and the words of encouragement you've provided. We are open and receptive to any further insights you might have to offer. We will respond to each question individually.
>
> Weakness 1: Wandering in the dynamic geometric network topology:
>
> Answer 1: Our approach incorporates a time-inhomogeneous Markov transition matrix, allowing us to dynamically adjust and adapt the regularized route of the mobilized server based on real-time information.
> Unlike a fixed random walk pattern, our method can intelligently modify the server's movement pattern to respond to changes in the combat environment. As the situation evolves and new data becomes available, the Markov transition matrix is updated in real-time, enabling the server to adjust its route and navigation strategy.
> This adaptability is particularly crucial in drastic combat scenarios, where the environment can change rapidly and unpredictably. By continuously updating the Markov transition matrix, our method ensures that the mobilized server remains agile and responsive to super dynamically changing conditions.
>
> Weakness 2: Stationary nodes
>
> Answer 2: The nodes are not stationary. They update their locations periodically to the server via satellite communications. Based on the real-time locations of the nodes and the vehicle as well as the status on the battlefield, the reachability graph can be calculated.
>
> Q1: On page 8, the authors claim RWSADMM’s O(\omega^{-1}) communication complexity is better than Per-FedAvg’s O(\omega^{-3/2}) complexity. This holds when \omega ranges from 0 to 1, but the authors seem not to discuss or define the range of the parameter.
>
> A1:  We can assume that \omega represents a small (less than 1) positive parameter similar to [22][25], which can be realized by controlling gradient magnitudes through common weight regularization techniques such as L1 or L2 regularization in the models. This could prompt the model to adopt lighter weights, resulting in reduced gradients, consequently yielding a smaller value for \omega.

---

> > ### Comment · Reviewer_J9bJ · 2023-08-12
> >
> > Thank you for your responses. They addressed some of my concerns about the applicability of the proposed framework.

---

### Official Review · Reviewer_46UP · 2023-07-06

**Soundness:** 3 good
**Presentation:** 3 good
**Contribution:** 2 fair
**Rating:** 6
**Confidence:** 4

**Summary:**

This paper proposes a new FL setting that involves a dynamically moving server among clients residing in different regions and receiving updates from clients of the selected region through wireless links. The proposed algorithm RWSADMM addresses personalization and data heterogeneity by minimizing the local proximity among neighbor nodes of each client. Additionally, the authors apply a first-order subgradient expansion for computational complexity reduction, which is useful for real-world applications. The algorithm is demonstrated to converge provably faster than baseline methods through theoretical analysis and experimental comparisons.

**Strengths:**

This paper introduces a novel asynchronous mobilizing federated learning setting that can be
utilized in realistic war situations where communication infrastructure is unavailable. Additionally,
it establishes a mathematical framework based on dynamic graph-based representation, allowing the server to move through the field and interact with various clients to update the neural network.

The proposed algorithm surpasses the baseline methods by a significant margin, demonstrating notably faster convergence rates, particularly in realistic non-IID scenarios.

The methodology employed in this study is a theoretically sound solution derived from the analysis of dynamic Markov Chain and Alternating Direction Method of Multipliers.

The proposed RWSADMM algorithm ensures convergence based on commonly assumed properties of the loss function. Furthermore, the communication complexity of RWSADMM is compared to existing methods like Per-FedAvg and APFL, providing theoretical evidence of its superior scalability.

In the proposed setting, both convex and non-convex models converge faster and exhibit higher accuracy performance compared to the baseline methods.


**Weaknesses:**

The proposed setting (mobilized federated learning) seems somewhat unrealistic. In a war situation where long-range communication is not supported, is there a need for federated learning? Considering that neural networks are large and transmitting them via tactical vehicles would be challenging, wouldn't it be better to transmit data for distributed learning or perform centralized learning on the server, rather than transmitting model updates?

The proposed Markov Chain does not seem to effectively represent the problem situation (mobilized federated learning). In real scenarios, nodes within the communication range of the communication vehicle should be selected, but the proposed algorithm does not consider the physical distances between nodes. Additionally, considering realistic situations like Figure 3, where nodes (walking soldiers) move and the topology keeps changing, the Markov Chain used in this paper cannot reflect spatial rearrangements of participating nodes.

I am concerned about privacy issues. In the proposed algorithm, each node needs to receive model updates from neighboring nodes to calculate the proximity term. However, this process can expose the data that the node possesses, as it can be extracted from the updates of other nodes. Furthermore, simultaneous reception of model updates from multiple nodes can result in communication overhead.

There is a lack of ablation study on the proposed components. There is no analysis on how well the proposed local proximity term resolves data heterogeneity, and no trade-off analysis on how much computational resources are saved through first-order expansion and the corresponding impact on performance.


**Questions:**

In the current problem, the server moves through a random walk. However, in practical situations, shouldn't the server actively modify its routing to reach unseen nodes?

Looking at the Experimental setup in Section 5, there are only 20 clients. Considering the trend in recently published Federated learning works, shouldn't the proposed algorithm be compatible with scenarios involving a larger number of clients?

In Figure 2, although PerAvg and APFL are FL works geared to data heterogeneity, their performances are significantly lower compared to FedAvg. Why does this discrepancy occur?

What are the confidence intervals for the reported performance in Figure 2 and Table 1? It is important to report confidence intervals, as performance can vary significantly  depending on how the spatial relationship between nodes is initialized.

The proposed setting (mobilized federated learning) seems somewhat unrealistic. In a war situation where long-range communication is not supported, is there a need for federated learning? Considering that neural networks are large and transmitting them via tactical vehicles would be challenging, wouldn't it be better to transmit data for distributed learning or perform centralized learning on the server, rather than transmitting model updates?

**Limitations:**

yes

---

> ### Author Rebuttal · Authors · 2023-08-09
>
> Thank you for your follow-up questions and comments. Below we address each question and minor questions in a point by point fashion.
>
> Weakness 1: Realism and Transmission burden:
>
> Answer 1: a) Realism of the Proposed Setting:
> Since we know that traditional federated learning may not be practical in such scenarios, the motivation for mobilized federated learning is to address the challenges posed by the constrained communication environment. In a war situation, communication infrastructure may be severely compromised or even non-existent. In contrast to centralized learning, which heavily relies on a stable and high-bandwidth connection to the server, mobilized federated learning leverages local communication within tactical vehicles. By adopting our mobilized federated learning, we strike a balance between model performance and maintaining data confidentiality. The decentralized nature of data ensures the continuity of learning even in adverse conditions, providing a vital advantage in the field.
> b) Transmission/Communication Burden:
> In the paper, we proved the improved communication complexity with other methods. Existing methods such as model compression and compiler compression can be readily combined with our method to improve the communication burden.
>
> Weakness 2: Restriction of the Markov Chain Representation:
>
> Answer 2: We employed a Time-inhomogeneous Markov Chain in our design, which allows for dynamic changes in the Markov Transition Matrix. This dynamic property enables the model to effectively depict the physical dynamical distance between nodes and account for the nodes' changing locations over time. In Figure 3, it can be observed that the nodes' locations undergo changes. Our primary objective was to develop a straightforward yet efficient method to capture the communication dynamics in response to dynamic topology within the network for communication.
> To achieve this, the server keeps receiving updated spatial information on the nodes via satellite communication. The random walk model we adopted operates in two iterative steps: Firstly, based on the locations of the nodes and the battle vehicle, the connectivity graph is updated. The subsequent stop for the battle vehicle is restricted to a few connected nodes that are reachable, effectively considering the constraints imposed by the vehicle's mobility and the reachability between the vehicle and nodes. Secondly, the random walk algorithm is executed to select one node from the reachable nodes. Thus, our random walk model incorporates considerations of the temporal and spatial correlation among nodes, further enhancing its performance in dynamic scenarios.
> While Figure 3 illustrates nodes' location changes with no drastic shifts, our model has been designed to cater to situations with more substantial topological variations. Our experiments demonstrate the model's robustness and effectiveness in practical scenarios with dynamic topologies, providing valuable insights into its applicability.
>
> Weakness 3: Privacy issues:
>
> Answer 3: To mitigate the privacy risks associated with the exchange of model updates among nodes, we can adopt specific measures to safeguard sensitive information. By implementing existing encrypted aggregation and privacy-preserving techniques, we can significantly reduce the exposure of individual data during the update calculation process.
>
> Weakness 4: Ablation study:
>
> Answer 4: In Section D.3 of the supplementary materials, we conducted a comprehensive hyperparameter study to assess the effectiveness of the local proximity term in addressing data heterogeneity through the parameter tuning of \beta. We further performed a thorough computational analysis to quantify the computational resource savings achieved by incorporating the first-order approximation, along with its impact on overall performance. We direct your attention to Table 2 enclosed within the attached PDF in the rebuttal. The findings depicted in the table reveal that the proposed first-order stochastic approximation yielded superior accuracy results while incurring less time costs. This outcome is well-founded since it obviates the need to solve the local subproblem to its optimal solution in every iteration. By mitigating the high risk of overfitting to local data and its associated computational overhead, our approach facilitates a more efficient global convergence process.
>
> Q1: Modify its routing to reach unseen nodes:
>
> A1: In our design, we have indeed taken this aspect into careful consideration. The server's movement is modeled as a generalized random walk which relies on a time-nonhomogeneous Markov matrix to determine the server's transitions. This dynamic feature allows us to modify the Markov matrix temporally, enabling the server to adapt its routing strategy as the situation evolves. By leveraging the time-nonhomogeneous Markov matrix, the server can effectively navigate through the network, reaching unseen nodes or avoiding hazardous areas in battlefields or buildings after a disaster.
>
> Q2: A larger number of clients in the experiment:
>
> A2:  In the main context, the majority of experiments were conducted using 20 clients. Nonetheless, to thoroughly assess the performance of RWSADMM with optimal configurations, additional experiments were conducted with 50 and 100 nodes. The empirical results from these experiments are provided in Appendix D.6.
>
> Q3: Discrepancy in the experimental result? Standard Deviation of the reported performance?
>
> A3: We observed that the performance of APFL appears to be lower compared to FedAvg in some settings. This discrepancy can be attributed to the handling of highly heterogeneous data across participant clients during one round of updates in the APFL algorithm. The experimental outcomes presented in Table 1 and Figure 2 have been updated in the response PDF. The calculation of standard deviations involves conducting ten iterations for each framework, indicating the confidence intervals.

---

> > ### Comment · Reviewer_46UP · 2023-08-15
> >
> > I thank the authors for their efforts in putting together their rebuttal. I am satisfied with the answers and explanations provided . I raise my score to 6.

---

### Author Rebuttal · Authors · 2023-08-09

We extend our gratitude to the reviewers for their insightful feedback, which has provided us an opportunity to clarify and refine our work. In response to the raised concerns, we present a comprehensive overview of our approach's enhancements and differentiators, as well as clarifications on critical aspects.

Oversimplification and Realism:
We have meticulously addressed the concern of oversimplification through the incorporation of a time-inhomogeneous Markov Probability Transition matrix (P(k)). This novel feature, in contrast to conventional fixed transition matrices, effectively captures dynamic movement patterns of nodes over time. The connectivity graph update and the subsequent random walk algorithm with accessible nodes collectively contribute to a more authentic portrayal of mobility patterns in real-world infrastructureless environments.

Distinguishing Asynchronous FL Framework:
Our RWSADMM framework stands apart from existing asynchronous FL methods, as we specialize in scenarios characterized by inadequate or non-existent network infrastructure. While traditional approaches assume certain infrastructure levels, our framework thrives in very constrained and infrastructureless environments, ensuring convergence without deterministic or stochastic bounded delay constraints which are often required in the existing asynchronous distributed learning. Furthermore, our method provably excels in challenging conditions through the integration of mobility patterns and dynamic environmental adjustments.

Adapting to Changing Topologies:
To address concerns about wandering topology, our approach integrates a time-inhomogeneous Markov transition matrix. This enables real-time adaptation of the mobilized server's route based on evolving information. By intelligently modifying the server's movement, our approach ensures responsiveness to rapidly changing combat environments, effectively addressing the issue of wandering topology.

Dynamic Nodes and Communication Burden:
Contrary to the assumption of stationary nodes, our nodes update positions via satellite communications, which enables calculation of the reachability graph based on real-time locations. We have demonstrated improved communication complexity and highlight the potential integration of model compression and compiler optimization techniques to alleviate communication burden.

Quantifying Complexities and Privacy Measures:
Regarding the clarification of \omega in our communication complexity claim, we clarify that it represents a small positive parameter, akin to existing references. To address privacy concerns, our approach is readily to work with existing encrypted aggregation and privacy-preserving techniques such as differential privacy to minimize data exposure during update calculations.

Ablation Study and Experimental Results:
Our supplementary materials contain a detailed hyperparameter study and trade-off analysis, which indicate the effectiveness of the local proximity term in resolving data heterogeneity. More results that are updated regarding the confidence intervals can be seen in the rebuttal PDF.

---

### Decision · Program_Chairs · 2023-09-21

**Decision:**

Accept (poster)

**Comment:**

The main challenge studied in the paper is to apply the methodology of Federated Learning (FL) to practical scenarios featuring isolated nodes with data heterogeneity, which can only be connected to the server through wireless links in an infrastructure-less environment. The paper proposes a mobilizing personalized FL approach, which aims to facilitate mobility and resilience. The main novelty of the proposed approach is to allow the server to make (random, but targeted) movement toward clients and formulates local proximity among their adjacent clients based on hard inequality constraints rather than requiring consensus updates or introducing bias via regularization methods. This of course integrated through an optimization method, hence introducing Random Walk Stochastic Alternating Direction Method of Multipliers (RWSADMM).

Most of the reviewers (also myself) are in favor of accepting the paper. After discussion the paper with the reviewers, I believe that the major concerns have been resolved. I recommend the authors to carefully apply all the excellent comments from the reviewers. Specifically the comments of Reviewer CJVn. I also think that the authors should rewrite the introduction in a way that it encompasses a more general set of scenarios (not just the ones related to warzones). The main reason that I am recommending the paper for acceptance is that I believe the proposed method will apply to more general settings (e.g. settings in robotics) and the idea of make the server mobile is a novel one.